# Understanding the Training Dynamics in Federated Deep Learning via Aggregation Weight Optimization

## Abstract

From the server's perspective, federated learning (FL) learns a global model by iteratively sampling a cohort of clients and updating the global model with the sum local gradient of the cohort. We find this process is analogous to mini-batch SGD of centralized training. In mini-batch SGD, a model is learned by iteratively sampling a batch of data and updating the model with the sum gradient of the batch. In this paper, we delve into the training dynamics in FL by learning from the experience of optimization and generalization in mini-batch SGD. Specifically, we focus on two aspects: *client coherence* (refers to sample coherence in mini-batch SGD) and *global weight shrinking regularization* (refers to weight decay in mini-batch SGD). We find the roles of the two aspects are both determined by the aggregation weights assigned to each client during global model updating. Thus, we use aggregation weight optimization on the server as a tool to study how client heterogeneity and the number of local epochs affect the global training dynamics in FL. Besides, we propose an effective method for **Fed**erated **A**ggregation **W**eight **O**ptimization, named as **FedAWO**. Extensive experiments verify that our method can improve the generalization of the global model by a large margin on different datasets and models.

## 1 Introduction

Federated Learning (FL) (McMahan et al., 2017; Li et al., 2020a; Wang et al., 2021; Lin et al., 2020; Li et al., 2022b) is a promising distributed optimization paradigm where clients' data are kept local, and a central server aggregates clients' local gradients for collaborative training. Although a lot of FL algorithms with deep neural networks (DNNs) are emerging in recent years (Lin et al., 2020; Chen & Chao, 2021a; Li et al., 2020b; Acar et al., 2020; Chen & Chao, 2021b), there are few works about the underlying training dynamics in FL with DNNs (Yan et al., 2021; Yuan et al., 2021), which hinders us to go further into the link between generalization and optimization in FL.

In the meanwhile, an interesting analogy exists between centralized mini-batch SGD and FL. The server-client training framework of FL (from the server perspective) learns a global model by iteratively sampling a cohort of clients and updating the global model with the sum local gradient of the cohort. While in centralized mini-batch SGD, a model is learned by iteratively sampling a mini-batch of data and updated by summing the corresponding gradients. In the analogy, the clients in FL refer to the data samples in mini-batch SGD, the cohort of clients refers to the mini-batch of data samples, and the communication round refers to the iteration step. The interesting analogy makes us wonder:

*Can we leverage the insights of mini-batch SGD to better understand the training dynamics in FL?*

Following this question and considering the key techniques in mini-batch SGD (as well as its generalization), in this paper, we focus on two aspects of training dynamics in FL: *client coherence* (refers to sample coherence in mini-batch SGD) and *global weight shrinking (GWS) regularization* (refers to weight decay in mini-batch SGD). Firstly, sample coherence explains how the relations between data samples affect the generalization of DNN models (Chatterjee, 2019; Chatterjee & Zielinski, 2020; Fort et al., 2019). As an analogy, here we extend the concept of sample coherence to the client case in FL with partial participation for studying the effect and training dynamics jointly caused by heterogeneous client data and local updates. Secondly, in a different line of works, weight decay methods (Lewkowycz & Gur-Ari, 2020; Zhang et al., 2018; Loshchilov & Hutter, 2018; Xie et al., 2020)—by decaying the weights of the model parameter in each iteration step—are the key

techniques in the mini-batch SGD based optimization to guard the generalization performance of deep learning tasks. We similarly examine the effects of weight decay in FL, in which we shrink the aggregated global model on the server in each communication round (i.e. global weight shrinking). Note that we take the server-side aggregation weight optimization as a tool framework to derive the insights of the training dynamics in FL. Though the idea of aggregation weight optimization was appeared in previous FL works to match similar peers in decentralized FL (Li et al., 2022a) or improve performances in FL with medical tasks (Xia et al., 2021), all prior works assume normalized aggregation weights of clients' models (i.e. $\gamma = 1$ in Equation 1), failing to dive into understand the FL's dynamics from the learned weights for further insights, e.g., identifying the significance of adaptive global weight shrinking.

**Specifically, our contributions are three-folded.**

- We first make an analogy between centralized mini-batch SGD and FL, in which it enables us to derive a principled tool framework to understand the training dynamics in FL, by leveraging the learnt aggregation weights a global-objective-consistent proxy dataset.
- As our main contribution, we identify some interesting findings (see below take-away) to unveil the training dynamics of FL, from the aspects of client coherence (cf. section 3) and global weight shrinking (cf. section 4) [1]. These insights are crucial to the FL community and can inspire better practical algorithm design in the future.
- We showcase the effectiveness of these insights, and devise a simple yet effective method FEDAWO, for server-side aggregation weight optimization (cf. section 5). It can perform adaptive global weight shrinking and optimize attentive aggregation weights simultaneously to improve the performance of the global model.

**We summarize our key take-away messages of the understandings as follows.**

- Our novel concept of client coherence undermines the training dynamics of FL, from the aspects of *local gradient coherence* and *heterogeneity coherence*.
  - Local gradient coherence refers to the averaged cosine similarities of clients' local gradients. A critical point (from positive to negative) exists in the curves of local gradient coherence during the training. The optimization quality of the initial phase (before encountering the point) matters: Assigning larger weights to more coherent clients in this period boosts the final performance.
  - Heterogeneity coherence refers to the distribution consistency between the global data and the sampled one (i.e. data distribution of a cohort of sampled clients) in each round. The value of heterogeneity coherence is proportional to the IID-ness of clients as well as the client participation ratio; the higher, the better. Increasing the heterogeneity coherence by reweighting the sampled clients could also improve the training performance.
- Global weight shrinking regularization effectively improves the generalization performance of the global model.
  - When the number of local epochs is larger, or the clients' data are more IID, a stronger global weight shrinking is necessary.
  - The magnitude of the global gradient (i.e. uniform average of local updates) determines the optimal weight shrinking factor. A larger norm of the global gradient requires stronger regularization.
  - In the late training of FL, where the global model is near convergence, the effect of global weight shrinking gradually saturates.
  - The effectiveness of global weight shrinking is stemmed from flatter loss landscapes of the global model as well as the improved local gradient coherence after the critical point.[2]

## 2  UNDERSTANDING FL VIA AN ANALOGY

The update step of FL[3] can be viewed as a manipulation of the received local models[3]:

$$\mathbf{w}_g^{t+1} = \gamma \cdot (\mathbf{w}_g^t - \eta_g \sum_{i=1}^{m} \lambda_i \mathbf{g}_i^t), \text{ s.t. } \gamma > 0, \lambda_i \geq 0, \|\boldsymbol{\lambda}\|_1 = 1, \tag{1}$$

where $\mathbf{w}_g^{t+1}$ denotes the global model of round $t + 1$, $\eta_g$ is the global learning rate, $m$ is the cohort size (i.e., the number of sampled clients), and $\mathbf{g}_i^t$ denotes the local accumulated model updates of

---

[1]For concision, in section 4 and section 3, if not mentioned otherwise, we all use CIFAR-10 as dataset and SimpleCNN as model. Experiments on more datasets and models are shown in section 5 and Appendix.

[2]Different from previous observations (w/o affecting the training dynamics), applying global weight shrinking results in a positive local gradient coherence after the critical point and the learning can benefit from it.

[3]We recommend the readers to check Appendix A for the preliminary of federated learning.

client $i$ starting from the received global model $\mathbf{w}_g^t$. We assume client $i$ trains the model for $E$ local epochs to derive $\mathbf{g}_i^t$. The set of parameters $\{\gamma, \boldsymbol{\lambda}\}$ in Equation 1 describes the model aggregation process in FL for one communication round, where we refer $\gamma$ as the weight shrinking factor and $\boldsymbol{\lambda}$ as the relative aggregation weights among the clients. FEDAVG is a special case by setting $\gamma = 1$, $\eta_g = 1$, and $\lambda_i = \frac{|\mathcal{D}_i|}{|\mathcal{D}|}, \forall i \in [m]$.

The formulation is the analogy to mini-batch SGD: our studied 1) client coherence and 2) global weight shrinking[4]—through respectively optimizing $\gamma$ and $\boldsymbol{\lambda}$ on a server proxy dataset—refer to weight decay and gradient coherence in mini-batch SGD, respectively. The considered proxy dataset has the same distribution as the global learning objective (i.e. a class-balanced case in this paper), thus the learned aggregation weights ($\{\gamma, \boldsymbol{\lambda}\}$) can reflect the contributions of clients and the optimal regularization factor towards this global objective. By connecting the learned weights and the training dynamics, we can know the roles of client heterogeneity and local updates in different learning periods. We use a relatively large proxy dataset (2000 samples in CIFAR-10 with balanced class distributions) for exploration purposes only in section 3 and section 4, while in section 5, we test our proposed FEDAWO on small proxy datasets (100 samples in CIFAR-10).

We review the insights of mini-batch SGD below and leverage them to better understand the training dynamics of FL later. For other related works, please refer to Appendix B.

**Remark 1 (Weight decay)** *Insights in mini-batch SGD can be detailed as*
- *The optimal weight decay factor is approximately inverse to the number of epochs, and the importance of applying weight decay diminishes when the training epochs are relatively long (Loshchilov & Hutter, 2018; Lewkowycz & Gur-Ari, 2020; Xie et al., 2020).*
- *The effectiveness of weight decay may be explained by the caused (1) larger effective learning rate (Zhang et al., 2018; Wan et al., 2021), and (2) flatter loss landscape (Lyu et al., 2022).*

**Gradient coherence.** Gradient coherence, or sample coherence, is a crucial technique for understanding the training dynamics of mini-batch SGD in centralized learning (Chatterjee, 2019; Zielinski et al., 2020; Chatterjee & Zielinski, 2020; Fort et al., 2019). The gradient coherence measures the pair-wise gradient similarity among samples. If they are highly similar, the overall gradient within a mini-batch will be stronger in certain directions, resulting in a dominantly faster loss reduction and better generalization (Chatterjee, 2019; Zielinski et al., 2020; Chatterjee & Zielinski, 2020).

**Remark 2 (Gradient coherence)** *The critical period exists in mini-batch SGD, captured by the gradient coherence: the low coherence in the early training phase damages the final generalization performance, no matter the value of coherence controlled later (Chatterjee & Zielinski, 2020).*

## 3 CLIENT COHERENCE

### 3.1 BASIC CONCEPT AND FORMULATION

Inspired by gradient coherence in mini-batch SGD, we study *client coherence* in FL, i.e., the *local gradient coherence* of clients' model updates in FL. In addition to this, the FL has another unique aspect of coherence, namely *heterogeneity coherence*.

**Local Gradient Coherence.** The gradient coherence in mini-batch SGD is at the data sample level. Analogously, we find similar conclusions at the client level in FL, where aggregating similar local gradients among clients will produce a stronger global gradient, improving generalization and vice versa. We deduce the gradient coherence in mini-batch SGD and local gradient coherence in FL under a unified equation below:

$$\Delta \mathcal{L}^t = \mathcal{L}(\mathbf{w}^t - \eta \mathbf{g}^t) - \mathcal{L}(\mathbf{w}^t) \approx -\eta \cdot \langle \mathbf{g}^t, \mathbf{g}^t \rangle = -\eta \cdot \langle \sum_{i=1}^m \lambda_i \mathbf{g}_i^t, \sum_{i=1}^m \lambda_i \mathbf{g}_i^t \rangle$$

$$= -\eta \cdot (\sum_{i=1}^m \lambda_i^2 \|\mathbf{g}_i^t\|^2 + \sum_{i,j,i \neq j} \lambda_i \lambda_j \langle \mathbf{g}_i^t, \mathbf{g}_j^t \rangle) = -\eta \cdot (\sum_{i=1}^m \lambda_i^2 \|\mathbf{g}_i^t\|^2 + \sum_{i,j,i \neq j} \lambda_i \lambda_j \cos(\mathbf{g}_i^t, \mathbf{g}_j^t) \|\mathbf{g}_i^t\| \|\mathbf{g}_j^t\|).$$

$$(2)$$

Equation 2 is a Taylor expansion of the loss function within one update. In mini-batch SGD, $t$ refers to the iteration step, $m$ is the batch size, and $\mathbf{g}_i^t$ is the gradient of a sample $i$ at iteration $t$.

---

[4]We use the word "shrink" instead of "decay" as it shrinks the global model rather than decaying the model by subtracting a decay term (used in traditional weight decay). Similar "shrink" can be found in Li et al. (2020c).

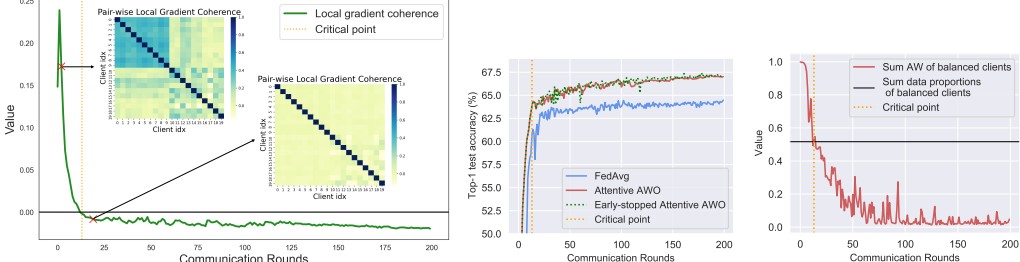

Figure 1: **Training dynamics of attentive AWO in terms of local gradient coherence.** Clients indexed 0-9 have balanced class distributions and 10-19 are imbalanced. **Left:** Local gradient coherence. **Middle and Right:** The performance and aggregation weights of attentive AWO and early-stopped attentive AWO.

Usually, there is no weighted averaging in a mini-batch, so $\forall i \in [m]$, $\lambda_i = 1$. In FL, $t$ refers to the communication round, $\mathbf{w}^t$ refers to the global model on the server at round $t$, $m$ is the cohort size, $\mathbf{g}_i^t$ denotes the local gradient of client $i$ at round $t$, and $\lambda_i$ is the aggregation weight of client $i$. In Equation 2, $\cos(\mathbf{g}_i^t, \mathbf{g}_j^t)$ means the cosine similarity of the two gradients, as $\langle \mathbf{g}_i^t, \mathbf{g}_j^t \rangle / \|\mathbf{g}_i^t\| \|\mathbf{g}_j^t\|$. Assuming all gradients have bounded norms that $\forall i$, $\|\mathbf{g}_i^t\| \leq \epsilon$. The cosine similarity among gradients indicates the coherence, from Equation 2, if the gradients have larger cosine similarity, it will have larger descent in the loss and improve the global generalization. In this paper, we focus on the local gradient coherence among clients during FL training. We borrow the cosine stiffness definition (Fort et al., 2019) to quantify the local gradient coherence in FL as follows.

**Definition 1** *The local gradient coherence of **two clients** $i$ **and** $j$ at round $t$ is defined by the cosine similarity of their local updates sent to the server, as:* $c_{(i,j)}^t = \cos(\mathbf{g}_i^t, \mathbf{g}_j^t)$.
*The overall local gradient coherence of **a cohort of clients** at round $t$ is defined by the weighted cosine similarity of all clients' local updates sent to the server, as:* $\mathbf{c}_{cohort}^t = \frac{1}{m} \sum_{i,j,i \neq j} \lambda_i \lambda_j \cos(\mathbf{g}_i^t, \mathbf{g}_j^t)$.

FL assumes multiple local epochs in each client, and clients usually have heterogeneous data; therefore, the local gradients of clients are usually almost orthogonal, that is to say, they have low coherence. This phenomenon is observed in Charles et al. (2021), but it did not dig deeper to examine the training dynamics of FL. In this paper, we calculate the local gradient coherence in each round and find a *critical point* exists in the process (Figure 1 and Figure 7).

**Heterogeneity Coherence.** Heterogeneity coherence refers to the distribution consistency between the global data and the sampled one (i.e. data distribution of a cohort of sampled clients) in each round. The value of heterogeneity coherence is proportional to the IID-ness of clients as well as the client participation ratio; the higher, the better. We define heterogeneity coherence as follows.

**Definition 2** *Assuming there are $N$ clients and the cohort size is $m$. For a given cohort of clients, the heterogeneity coherence is* $\text{sim}(\mathcal{D}_{cohort}, \mathcal{D})$*, where* $\mathcal{D}_{cohort} = \sum_{i \in [m]} \lambda_i \mathcal{D}_i, \mathcal{D} = \sum_{j=1}^{N} \lambda_j \mathcal{D}_j$ *and* sim *is the similarity of two data distributions.*

### 3.2 ATTENTIVE AGGREGATION WEIGHT OPTIMIZATION AND TRAINING DYNAMICS

Vanilla FEDAVG only considers data sizes in clients' aggregation weights $\boldsymbol{\lambda}$. However, client heterogeneity is also crucial. Clients with different heterogeneity degrees have different importance in client coherence, thus playing different roles in training dynamics. A three-node toy example is shown in Figure 11 in Appendix. The optimal $\boldsymbol{\lambda}$ is off the data-sized when clients have the same data size but different heterogeneity degrees. To study client coherence further, we propose attentive aggregation weight optimization (attentive AWO) to learn the optimal aggregation weights (i.e. $\boldsymbol{\lambda}$) on a proxy dataset. By connecting the optimal weights and the client coherence, we can know the roles of different clients in different learning periods. Attentive AWO conducts the model updates as in Equation 1, where $\{\gamma, \boldsymbol{\lambda}\}$ is defined as

$$\{\gamma = 1, \boldsymbol{\lambda} = \boldsymbol{\lambda}^*\}, \text{ where } \boldsymbol{\lambda}^* = \arg\min_{\boldsymbol{\lambda}} \mathcal{L}_{proxy}(\mathbf{w}_g^t - \eta_g \sum_{i=1}^{m} \lambda_i \mathbf{g}_i^t), \text{ s.t. } \lambda_i \geq 0, \|\boldsymbol{\lambda}\|_1 = 1. \quad (3)$$

**1) Critical point exists in terms of local gradient coherence.** To study the role of client data heterogeneity in local gradient coherence, we experiment on both balanced and imbalanced clients. Clients indexed 0-9 have balanced class distributions and 10-19 are imbalanced (the data distribution of clients is in Figure 12 of Appendix). We set full participation to engage all clients in each round. The results are demonstrated in Figure 1. It is found that *in the first couple of rounds, the coherence*

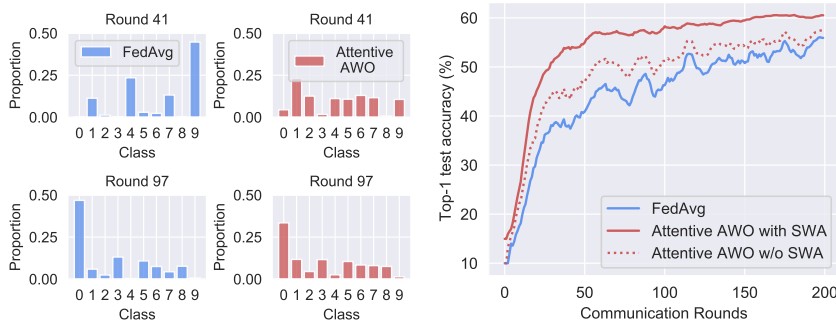

Figure 2: **Left:** Heterogeneity coherence of class distribution within a cohort. **Right:** Test accuracy curves.

*is dominant and positive, thus the test accuracy arises dramatically, and most generalization gains happen in this period.* The critical point is at the round that the coherence is near zero. After the critical point, the test accuracy gain is marginal, and the coherence is kept negative but close to zero.

**2) Assigning larger weights to clients with larger coherence before the critical point can improve overall performance.** From the Left of Figure 1, it is clear that before the critical point, the coherence among balanced clients is much more dominant than others, and it reveals that *clients with more balanced data have more coherent gradients*.[5] Intuitively, if we assign the balanced clients with larger weights before the critical point, it can boost generalization: in Equation 2, larger $\lambda$ assigning to clients with larger cosine similarity can reduce the loss more significantly.

Interestingly, we find attentive AWO proves our hypothesis: it raises the weights of balanced clients highly in the first few rounds; especially in the first two rounds, it nearly assigns all weights to the balanced clients. From the Left of Figure 1, after the point, clients have negative averaged coherence, and their mutual coherence is uniformly small, so the coherence gap between the balanced and imbalanced clients is not obvious. In this scenario, we believe that *the coherence of clients matters only before the critical point.* To verify this, we adopt early stopping near the critical point when conducting attentive AWO, that is, before the stopping round, we use learned weights to generate global models and then use data-sized weights afterwards. Results show that *the early-stopped attentive AWO has comparable performance after the critical point, indicating the training period before local gradient coherence reaches zero is much more vital.* This phenomenon is insightful for the community to design effective algorithms for learning critically in early training.

**3) Improving heterogeneity coherence within a cohort can boost performance.** In the scenarios with partial client participation in each round, the selected clients have inconsistent sum objective with the global objective, in other words, the heterogeneity coherence is low (Definition 2). More specifically, for the class-imbalanced setting, due to the local heterogeneous data, the sum local data of the randomly selected participating clients are extremely class-imbalanced whilst the sum data of all clients are class-balanced. We notice that *attentive AWO can improve heterogeneity coherence by dynamically adjusting the AW among clients.* We visualize the weighted class distributions within a cohort in Figure 2, and it shows attentive AWO learns weights to make the class distributions more balanced. The test accuracy curves demonstrate the significant performance gain compared with FEDAVG, and we notice that attentive AWO with SWA[6] performs better by seeking a more generalized minimum in the aggregation weight hyperplane. Due to the space limit, we include more analysis about client coherence in subsection C.1 of Appendix C.

## 4 GLOBAL WEIGHT SHRINKING

### 4.1 GLOBAL WEIGHT SHRINKING AND ITS IMPACTS ON OPTIMIZATION

As stated in Equation 1, setting $\gamma < 1$ results in the global weight shrinking regularization. Table 1 and the accuracy curves of different $\gamma$ in Figure 3 report the results on CIFAR-10 with different $\gamma$. It can be observed that the *global weight shrinking may improve generalization, depending on the choice of $\gamma$.* Moreover, an optimal $\gamma$ may exist, and a different value (either smaller or larger) will

---

[5]This also reveals why FL performs better in IID settings than NonIID: the clients' gradients in IID settings are more coherent, but the ones in the NonIID usually diverge.

[6]Stochastic Weight Averaging (SWA) (Izmailov et al., 2018) is an effective technique to make simple averaging of multiple points along the trajectory of optimization, with a cyclical learning rate. It leads to better generalization performance as well as a flatter minimum in DNNs.

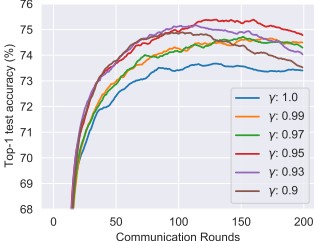 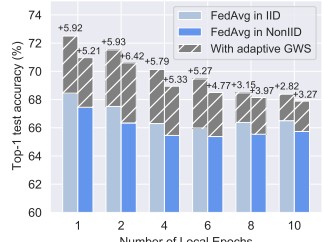 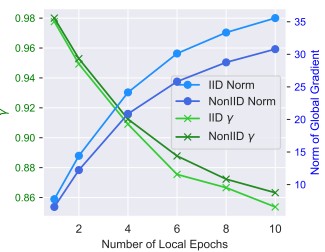

Figure 3: Test accuracy curves with different values of $\gamma$.

Figure 4: Test accuracy gains of adaptive GWS.

Figure 5: The optimal $\gamma$ and the norm of global gradient.

Table 1: Impact of fixed $\gamma$ across different architectures in both IID and NonIID settings ($E = 2$).

| | IID ($\alpha = 100$) | | | | | | NonIID ($\alpha = 1$) | | | | | |
| Model | 1.0 | 0.99 | 0.97 | 0.95 | 0.93 | 0.9 | 1.0 | 0.99 | 0.97 | 0.95 | 0.93 | 0.9 |
| --- | --- | --- | --- | --- | --- | --- | --- | --- | --- | --- | --- | --- |
| SimpleCNN | 65.53 | 67.60 | 69.20 | 69.52 | **70.16** | 69.83 | 65.58 | 67.04 | 68.36 | 68.66 | **69.28** | 68.93 |
| AlexNet | 74.16 | 74.80 | **75.54** | 75.24 | 75.25 | 75.03 | 73.56 | 73.83 | 74.37 | **74.45** | 74.40 | 74.24 |
| ResNet8 | 75.51 | 76.64 | 76.80 | **77.87** | 76.80 | 76.74 | 75.02 | 76.06 | 75.73 | **77.00** | 75.04 | 75.31 |

have inferior performance. Obviously, *if $\gamma$ is inappropriate, it will lead to performance degradation in the late training period*. In particular, for smaller $\gamma$, the degradation comes earlier, even if before the model reaches convergence. This observation is consistent with weight decay in centralized training (Remark 1). More results about the fixed $\gamma$ can be found in Table 7 in Appendix.

## 4.2 ADAPTIVE GLOBAL WEIGHT SHRINKING AND TRAINING DYNAMICS

To study the optimal $\gamma$ and the underlying impact factor of optimal $\gamma$, we realize adaptive global weight shrinking (adaptive GWS) on the proxy dataset. The proxy dataset represents the global learning objective, thus the learned $\gamma$ is the optimal value in each round towards this objective. Adaptive GWS adopts the update in Equation 1, where $\{\gamma, \lambda_i\}$ is defined as

$$\{\gamma = \gamma^*, \lambda_i = \frac{|\mathcal{D}_i|}{|\mathcal{D}|}\}, \text{ where } \gamma^* = \arg\min_{\gamma} \mathcal{L}_{proxy}(\gamma \cdot (\mathbf{w}_g^t - \eta_g \sum_{i=1}^{m} \frac{|\mathcal{D}_i|}{|\mathcal{D}|} \mathbf{g}_i^t)), \text{ s.t. } \gamma \geq 0. \quad (4)$$

Here, we fix $\boldsymbol{\lambda}$ as the data-sized weights. The experimental results are shown in Figure 4. Adaptive GWS can improve the performance of FEDAVG by a large margin in both IID and NonIID settings. We observe that adaptive GWS is more beneficial when the number of local epochs is small.

**1) Local epochs and client heterogeneity affect the optimal $\gamma$.** From the figure of Figure 5, *optimal $\gamma$ decreases when the local epoch increases or data become more IID, causing stronger weight shrinking regularization*. We consider this due to the balance between optimization and regularization. For a larger global gradient, it requires a stronger regularization term. We expand the update as

$$\mathbf{w}_g^{t+1} = \gamma(\mathbf{w}_g^t - \eta_g \mathbf{g}_g^t) = \mathbf{w}_g^t - \gamma\eta_g \mathbf{g}_g^t - (1 - \gamma)\mathbf{w}_g^t. \quad (5)$$

We refer $(1 - \gamma)\mathbf{w}_g^t$ as the pseudo-gradient of global weight shrinking regularization and $\gamma\eta_g \mathbf{g}_g^t$ is the global averaged gradient. We notice that *the changes of optimal $\gamma$ is due to the changes of the global gradient*. As shown in the right blue Y-axis of Figure 5, the norm of global gradient $\|\gamma\eta_g \mathbf{g}_g^t\|$ increases when the local epoch increases and data become IID. The larger global gradients have smaller optimal $\gamma$ (shown in the left green Y-axis) to produce a larger weight shrinking pseudo-gradient $\|(1 - \gamma)\mathbf{w}_g^t\|$ to regularize the optimization. More results regarding how heterogeneity affects the optimal $\gamma$ can be found in Figure 13 in Appendix.

**2) Optimal $\gamma$ increases in the late training.** As we discussed in subsection 4.1, GWS with smaller fixed $\gamma$ will cause performance degradation in the late training. We reckon the phenomenon is because the model is near convergence and the norm of the global gradient is decaying in the late training. If $\gamma$ is fixed, the regularization pseudo-gradient will not decay and dominate the optimization. Thus, in the late training, regularization should be decaying along with the model reaching convergence (smaller global gradient). Figure 6 further verifies our explanation that it shows the optimal $\gamma$ is decaying in the late period. *While the norm of the global gradient is decaying, adaptive GWS learns a rising $\gamma$ to keep the GWS pseudo-gradient decay proportionally*. As a result, the ratio of two gradient terms remains steady at around 19 to maintain the balance between optimization and regularization.

**3) The mechanisms behind adaptive GWS.** We provide two general understandings of how adaptive GWS changes the model parameter. We also study why adaptive GWS can improve generalization.

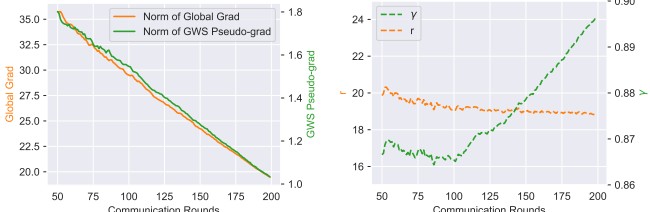 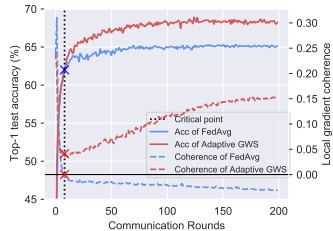

Figure 6: **Left:** Norm of two gradients in adaptive GWS. **Right:** The optimal $\gamma$ and $r$ in adaptive GWS, where $r$ is the ratio of the global gradient and the regularization pseudo-gradient.

Figure 7: Training dynamics of adaptive GWS in terms of local gradient coherence.

- **General understanding.**
  - **Scale invariance.** Adaptive GWS learns a dynamic shrinking factor $\gamma$ in each round to shrink the global model's parameter. The reason why the shrunk networks still works comes from the fact of the scale invariance property of DNNs (Li et al., 2018; Dinh et al., 2017; Kwon et al., 2021). It means that due to the non-linearity of activation functions or the normalization layer in DNNs, if a factor rescales the model weights, the function of the model remains similar or the same. We show an intuitive understanding of scale invariance on the Left figure of Figure 8. The final models are rescaled by $\gamma$, and the loss function of the adaptive GWS's final model remains similar while the FEDAVG's final model even has a smaller loss when $\gamma < 1$.
  - **Small model parameters.** *The shrinking effect in each round can result in smaller model parameters of the final global model*. The parameter weight histogram is demonstrated in the Middle figure of Figure 8. The final model of adaptive GWS has more model weights close to zero, nearly twice as many as FEDAVG.
- **Why adaptive GWS can improve generalization.**
  - **Flatter loss landscapes.** One perspective of explaining the generalization of NN is the flatness of the loss landscape. Previous researches find that flatter curvature in loss landscape can indicate better generalization (Fort & Jastrzebski, 2019; Foret et al., 2020; Li et al., 2018). In Lyu et al. (2022), they find weight decay of mini-batch SGD can result in flatter landscapes in DNNs with normalization layers. We also observe the similar phenomenon that *adaptive GWS improves generalization by seeking flatter minima in FL*, shown in the Right figure of Figure 8. Along the hessian eigenvector with maximal eigenvalue, it is clear that *the model of adaptive GWS has a flatter landscape than FEDAVG, and it also has a smaller loss*. Additionally, we also use other flatness metrics based on hessian eigenvalues to compare the loss landscapes during training in Figure 14 in Appendix, and these metrics also show that adaptive GWS can result in flatter curvature with better generalization.
  - **Improving local gradient coherence after the critical point.** In section 3, we find that when $\gamma = 1$, there exists a critical point where local gradient coherence turns positive to negative, and after the critical point, the generalization gain is marginal. However, we find *if we adopt adaptive GWS, the local gradient coherence is still positive after the critical point, and the model can still benefit from the coherent gradients*. We demonstrate the local gradient coherence and test accuracies in one figure in Figure 7. Before the critical point, the vanilla FEDAVG and adaptive GWS both have high gradient coherence, so the accuracies rise equally fast. However, after the critical point, the coherence of FEDAVG goes down below zero. Therefore, the generalization has little performance gain afterwards, and the optimization is near saturation. On the contrary, after the critical point, adaptive GWS keeps the coherence above zero, and the global model still benefits from having a larger performance gain beyond FEDAVG. This shows that the *shrinking regularization benefits from the long-time optimization after the critical point by making clients' local gradients more coherent*.

**4) The relation between adaptive GWS and local weight decay.** Our proposed adaptive GWS can cause weight regularization from the global perspective, which is analogous to weight decay in mini-batch SGD. Importantly, GWS has a sparse regularization frequency that only changes the model weight in each round, and as a result, we find GWS has stronger regularization each time. In GWS, $1 - \gamma$ is near 0.1, whereas the factor of weight decay is about $10^{-4}$. The two methods are not conflicted in FL, and we conduct experiments on implementing weight decay in the local SGD solver and global weight shrinking on the server simultaneously in Table 2. It is shown that *adaptive GWS is compatible with local weight decay and can further improve performance.* Local weight decay relies on the hyperparameter of the decay rate, and it needs hyperparameter tuning to search for the most appropriate value in every setting. *Instead, adaptive GWS is hyperparameter-free and effective*:

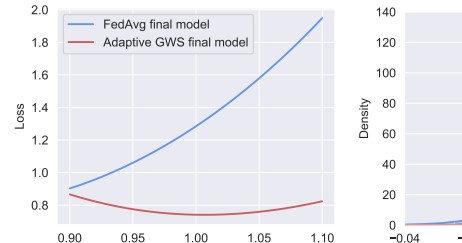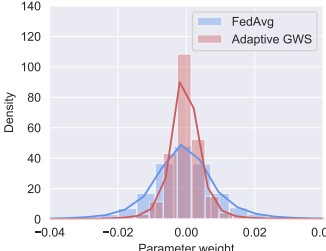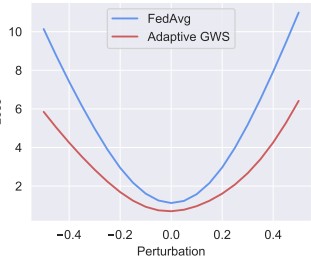

Figure 8: **General understanding of adaptive GWS. Left:** Scale invariance property of NN indicates that if the network is rescaled by $\gamma$, the function of the model remains similar. **Middle:** The histogram of final models' weights shows that adaptive GWS makes more model weights close to zero, nearly twice as many as FEDAVG. **Right:** The Loss landscape is perturbed based on the Top-1 Hessian eigenvector of the final models, which shows that the model with adaptive GWS has flatter curvature and smaller loss.

Table 2: Adaptive GWS with different local weight decay factors ($E = 2$).

| Local weight decay | IID ($\alpha = 100$) | | | | | NonIID ($\alpha = 1$) | | | | |
|---|---|---|---|---|---|---|---|---|---|---|
| | 0 | 5e-5 | 1e-4 | 5e-4 | 1e-3 | 0 | 5e-5 | 1e-4 | 5e-4 | 1e-3 |
| FEDAVG | 66.43 | 66.20 | 66.45 | 67.51 | 68.66 | 65.35 | 65.19 | 65.77 | 66.37 | 67.4 |
| Adaptive GWS | **71.47** | **71.36** | **71.35** | **71.44** | **71.54** | **70.31** | 69.93 | **70.44** | **70.47** | 69.99 |
| $\gamma$ of Adaptive GWS | 0.9472 | 0.9477 | 0.948 | 0.9493 | 0.953 | 0.9492 | 0.9499 | 0.9505 | 0.9529 | 0.9561 |

Table 3: The performance with different percentages of corrupted clients (IID, $E = 3$)

| Method | 25% | 50% | 75% |
|---|---|---|---|
| FEDAVG | 63.40 | 61.14 | 58.21 |
| FEDDF | 68.73 | 66.94 | 66.07 |
| SERVER-FT | 63.61 | 61.24 | 58.36 |
| FEDAWO | 67.87 | 66.67 | 63.91 |
| FEDAWO (SWA) | 68.04 | 66.95 | 65.51 |

Table 4: The Performance of compared methods with different model architectures ($\alpha = 1, E = 1$).

| Model | FEDAVG | FEDAWO | FEDAWO (SWA) |
|---|---|---|---|
| ResNet20 | 74.11 | 78.72 | 78.64 |
| ResNet56 | 74.22 | 78.93 | 79.08 |
| ResNet110 | 74.50 | 78.11 | 79.19 |
| WRN56_4 | 78.67 | 79.61 | 80.70 |
| DenseNet121 | 85.13 | 86.50 | 87.06 |

according to the results, no matter what the local weight decay factors and client heterogeneity, it can adaptively set $\gamma$ to maximize the benefit of weight regularization. Another interesting finding is that *adaptive GWS can balance the regularization and optimization dynamically: the learned $\gamma$ is adaptively adjusted in different local weight decay factors*. When the local weight decay is stronger, it will have larger $\gamma$, which means weaker GWS regularization. Due to the space limit, we include more analysis about global weight shrinking in subsection C.2 of Appendix C.

## 5 FEDERATED AGGREGATION WEIGHT OPTIMIZATION: FEDAWO

Based on the above understandings, we propose FEDAWO which combines the adaptive GWS and attentive AWO to optimize $\gamma$ and $\boldsymbol{\lambda}$ simultaneously, defined as

$$\{\gamma = \gamma^*, \boldsymbol{\lambda} = \boldsymbol{\lambda}^*\}, \text{ where } \boldsymbol{\gamma}^*, \boldsymbol{\lambda}^* = \arg\min_{\gamma,\boldsymbol{\lambda}} \mathcal{L}_{proxy}\gamma \cdot (\mathbf{w}_g^t - \eta_g \sum_{i=1}^{m} \lambda_i \mathbf{g}_i^t),, \text{ s.t. } \gamma \geq 0, \lambda_i \geq 0, \|\boldsymbol{\lambda}\|_1 = 1. \tag{6}$$

The optimization of $\gamma$ and $\boldsymbol{\lambda}$ is non-trivial. First of all, it is unclear how to treat the normalization layers. The models with normalization layers contain buffers to calculate the running mean and variance of training data, and naively taking buffers as model parameters and multiplying them with learned $\gamma$ will impede optimization. As our solution, we do not aggregate the buffers; instead, we update the buffers on the proxy dataset during AW optimization. Even though the proxy dataset is relatively small (e.g. 100 images in total), the updated buffers still work well to enable the global model with good generalization. Additionally, it is challenging to incorporate with SWA (Izmailov et al., 2018) for better generalization performance, as a joint optimization of $\boldsymbol{\lambda}$ and $\gamma$ with SWA has poor performance due to the sensitivity of $\gamma$ on stochastic averaging. To solve this, we adopt an alternative two-stage strategy for SWA variant (implementing it in a reversed order also works), where we first fix $\boldsymbol{\lambda}$ and optimize $\gamma$, then we use the learned $\gamma$ and fix it to optimize $\boldsymbol{\lambda}$ with SWA.

**Experiments.** We conduct experiments to verify the effectiveness of FEDAWO. Due to the page length limitation, for more information, please refer to Appendix D (details of FEDAWO) and

Table 5: **Top-1 test accuracy (%) achieved by compared FL methods and FedAWO on three datasets with different model architectures** ($E = 3$). Blue/**bold** fonts highlight the best baseline/our approach.

| Dataset | FashionMNIST | | | | CIFAR-10 | | | | CIFAR-100 | | | |
|---|---|---|---|---|---|---|---|---|---|---|---|---|
| NonIID ($\alpha$) | 100 | | 0.1 | | 100 | | 0.1 | | 100 | | 0.1 | |
| Model | MLP | LeNet | MLP | LeNet | CNN | ResNet | CNN | ResNet | CNN | ResNet | CNN | ResNet |
| FEDAVG | 89.29 | 90.54 | 85.11 | 88.08 | 65.78 | 74.57 | 60.13 | 46.04 | 25.74 | 27.49 | 27.74 | 24.92 |
| FEDPROX | 87.68 | 89.77 | 84.33 | 87.01 | 67.66 | 68.51 | 60.48 | 48.84 | 9.49 | 27.15 | 12.52 | 23.73 |
| FEDDYN | 88.47 | 89.92 | 77.68 | 72.68 | 66.1 | 76.62 | 41.53 | 35.77 | 24.44 | 32.18 | 22.67 | 29.00 |
| FEDDF | 86.16 | 89.09 | 78.48 | 85.90 | 69.60 | 77.36 | 57.38 | 54.09 | 28.52 | 27.42 | 24.52 | 23.10 |
| FEDBE | 86.22 | 89.14 | 79.12 | 85.96 | 69.88 | 77.94 | 59.84 | 52.86 | 28.38 | 27.73 | 25.41 | 23.74 |
| SERVER-FT | 89.09 | 90.56 | 85.71 | 88.10 | 66.83 | 74.73 | 60.43 | 47.59 | 25.37 | 26.14 | 24.33 | 23.03 |
| **FEDAWO** | **88.51** | **90.66** | **86.30** | **88.26** | **70.17** | **80.46** | **62.46** | **52.83** | **32.51** | **33.17** | **32.30** | **24.84** |
| **FEDAWO (SWA)** | **88.27** | **90.51** | **86.89** | **88.18** | **69.9** | **79.55** | **62.12** | **57.08** | **32.39** | **33.17** | **32.27** | **25.31** |

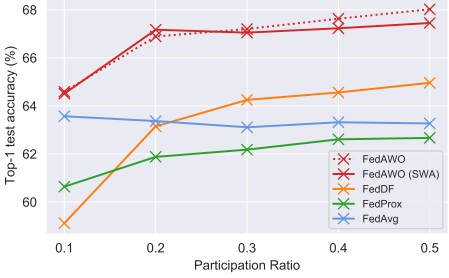

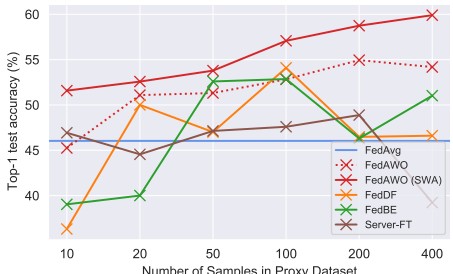

Figure 9: The performance with different participation ratios ($\alpha = 1$, $E = 3$).

Figure 10: The performance with different sizes of the proxy dataset ($\alpha = 0.1$, $E = 3$).

Appendix E (implementation details). We mainly compare FEDAWO with other server-side methods, i.e. FEDDF (Lin et al., 2020) and FEDBE (Chen & Chao, 2021a), that also require a proxy dataset for additional computation. These two methods conduct ensemble distillation on the proxy data to transfer knowledge from clients' models to the global model. We add SERVER-FT as a baseline for simply finetuning global models on the proxy dataset. Besides, we implement client-side algorithms FEDPROX (Li et al., 2020b) and FEDDYN (Acar et al., 2020).

**Results.** *Different datasets:* As in Table 5, FEDAWO outperforms baselines across different datasets and models in both IID and NonIID settings. Compared with FEDDF, FEDBE and SERVER-FT, FEDAWO can better utilize the proxy dataset. *Different participation ratios:* From Figure 9, FEDAWO performs well under partial participation. *Different sizes and distributions of proxy dataset:* In Figure 10, the server-side baselines are sensitive to the size of the proxy dataset that too small or too large proxy set will cause overfitting. However, FEDAWO is also effective under an extremely tiny proxy set and benefits more from a larger proxy set due to accurate aggregation weight optimization. We report the results of different distributions of the proxy dataset in subsection C.4 of Appendix C, which show that FedAWO still works when there exists a distribution shift between the proxy dataset and the gloabl data distribution of clients. *Different architectures:* We test FEDAWO across wider and deeper ResNet and other architecture, such as DenseNet (Huang et al., 2017), in the Table 4. It shows FEDAWO is effective across different architectures, and it performs well even when the network goes deeper or wider. *Robustness against corrupted clients:* Another advantage of FEDAWO is that it can filter out corrupted clients by assigning them lower weights. We generate corrupted clients by swapping two labels in their local training data. As in Table 3, FEDAWO has great performance even corrupted clients exist, and it is as robust as the ensemble distillation methods, like FEDDF, when using the same proxy dataset.

## 6 CONCLUSION

We introduce a novel global optimization framework to understand the training dynamics in FL, by making an analogy to mini-batch SGD in centralized learning. Leveraging the framework, we identify some interesting findings to unveil the FL training dynamics in terms of global weight shrinking regularization and client coherence. Based on the findings, we devise a simple but effective method FEDAWO, for server-side aggregation weight optimization. It can realize both adaptive global weight shrinking and attentive aggregation weights to better improve the generalization of the global model.

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

# Appendix

In this appendix, we provide details omitted in the main paper and more experimental results and analyses.

- Appendix A: preliminary of federated learning (cf. section 1 and section 2 of the main paper).
- Appendix B: related works (cf. section 1 of the main paper).
- Appendix C: more experimental results and analyses (cf. section 3, section 4 and section 5 of the main paper).
- Appendix D: additional details of FEDAWO (cf. section 3, section 4 and section 5 of the main paper).
- Appendix E: details of experimental setups (cf. section 5 of the main paper).

## A   PRELIMINARY OF FEDERATED LEARNING

Federated learning usually involves a server and $n$ clients to jointly learn a global model without data sharing, which is originally proposed in (McMahan et al., 2017). Denote the set of clients by $\mathcal{S}$, the labeled data of client $i$ by $\mathcal{D}_i = \{(x_j, y_j)\}_{j=1}^{N_i}$, and the parameters of the current global model by $\mathbf{w}_g^t$. FL starts with **client training** in parallel, initializing each clients' model $\mathbf{w}_i^t$ with $\mathbf{w}_g^t$.

FL is more communication-efficient than the conventional distributed training, that it assumes the clients train the models for epochs (the full data) instead of iterations (the mini-batch data) between the communications to the server. **The number of local epochs is denoted as $E$.**

In each local epoch, clients conduct SGD update with a local learning rate $\eta_l$, each SGD iteration shows as

$$\text{Client training:} \qquad \mathbf{w}_i^t \leftarrow \mathbf{w}_i^t - \eta_l \nabla \ell(B_k, \mathbf{w}_i^t), \text{ for } k = 1, 2, \cdots, K, \qquad (7)$$

where $\ell$ is the loss function and $B_k$ is the mini-batch sampled from $\mathcal{D}_i$ at the $k$th iteration. After the client local updates, the server samples $m$ clients for aggregation. The client $i$'s pseudo gradient of local updates is denoted as $\mathbf{g}_i^t = \mathbf{w}_g^t - \mathbf{w}_i^t$. Then, the server conducts FEDAVG to aggregate the local updates into a new global model.

$$\text{Weighted Model aggregation:} \qquad \mathbf{w}_g^{t+1} = \mathbf{w}_g^t - \sum_{i=1}^{m} \lambda_i \mathbf{g}_i^t, \ \lambda_i = \frac{|\mathcal{D}_i|}{|\mathcal{D}|}, \forall i \in [m]. \qquad (8)$$

With the updated global model $\mathbf{w}_g^{t+1}$, it then starts the next round of **client training**. The whole procedure of FL therefore iterates between Equation 7 and Equation 8, **for $T$ communication rounds**. We assume the sum of clients' data as $\mathcal{D} = \bigcup_{i \in \mathcal{S}} \mathcal{D}_i$. The IID data distributions of clients refer to that each client's distribution $\mathcal{D}_i$ is IID sampled from $\mathcal{D}$. However, in practical FL scenarios, **heterogeneity** exists among clients that their data are **NonIID** with each other. Each client may have different data distributions in the input (e.g. image distribution) or output (e.g. label distribution).

In this paper, we study how **the number of local epochs $E$** and **the clients' data heterogeneity** affect the training dynamics in terms of client coherence and global weight shrinking.

## B   RELATED WORKS

### B.1   MODEL AGGREGATION IN FEDERATED LEARNING

**Model aggregation in federated learning.** Model aggregation weights should be calibrated under asynchronous local updates. FEDNOVA (Wang et al., 2020b) is proposed to tackle the objective inconsistency problem caused by asynchronous updates, it theoretically shows that the convergence will be improved if the aggregation weights are normalized by the numbers of local iterations. However, it does not take the heterogeneity degree of clients into account, which is also a key factor that affects the generalization of the global model. In Chen & Chao (2021a), the authors point out that due to heterogeneity, the best performing model will shift way from FEDAVG, but they do not give insights on how to adjust aggregation weight to approximate the best model, they use Bayesian ensemble distillation method to prove the generalization of the global model instead. To solve the misalignment of neurons in FL with NN, FEDMA (Wang et al., 2020a) is proposed: FEDMA constructs the shared global model layer-wise by matching and averaging hidden elements

with similar features extraction signatures. Besides, optimal transport (Kantorovich, 2006) can be adopted in layer-wise neuron alignment in the process of model fusion (Singh & Jaggi, 2020). These previous works improve the global model performance by layer-wise alignment, but they are complex and computation-expensive, and they can not be applied under the traditional weighted aggregation scheme. What's more, distillation (Hinton et al., 2015; Zhu et al., 2021; Lin et al., 2020) can enable robust knowledge transfer and ensemble distillation can be used to finetune the global model for better generalization (Lin et al., 2020; Chen & Chao, 2021a).

### B.2    GENERALIZATION AND TRAINING DYNAMICS OF NEURAL NETWORK

**Loss landscape of neural networks and generalization.** Neural networks (NN) are highly non-convex and over-parameterized, and visualizing the loss landscape of NN (Li et al., 2018; Vlaar & Frankle, 2021) helps understand the training process and the properties of minima. There are mainly two lines of works about the loss landscape of NN. The first one is the linear interpolation of neural network loss landscape (Vlaar & Frankle, 2021; Garipov et al., 2018; Draxler et al., 2018), it plots linear slices of the landscape between two networks. In linear interpolation loss landscape, mode connectivity (Draxler et al., 2018; Vlaar & Frankle, 2021; Entezari et al., 2022) is referred to as the phenomenon that there might be increasing loss on the linear path between two minima found by SGD, and the loss increase on the path between two minima is referred to as (energy) barrier. It is also found that there may exist barriers between the initial model and the trained model (Vlaar & Frankle, 2021). The second line concerns the loss landscape around a trained model's parameters (Li et al., 2018). It is shown that the flatness of loss landscape curvature can reflect the generalization (Foret et al., 2020; Izmailov et al., 2018) and top hessian eigenvalues can present flatness (Yao et al., 2020; Jastrzębski et al., 2018). Networks with small top hessian eigenvalues have flat curvature and generalize well. Previous works seek flatter minima for improving generalization by implicitly regularizing the hessian (Foret et al., 2020; Kwon et al., 2021; Du et al., 2021).

**Critical learning period in training neural networks.** Jastrzebski et al. (2019) found that the early phase of training of deep neural networks is critical for their final performance. They show that a break-even point exists on the learning trajectory, beyond which the curvature of the loss surface and noise in the gradient are implicitly regularized by SGD. They also found that using a large learning rate in the initial phase of training reduces the variance of the gradient and improves generalization. In FL, Yan et al. (2021) discovers the early training period is also critical to federated learning. They reduce the quantity of training data in the first couple of rounds and then recover the training data, and it is found that no matter how much data are added in the late period, the models still cannot reach a better accuracy. However, it did not further study the role of client heterogeneity in the critical learning period while we examine it by local gradient coherence.

### B.3    FEDERATED HYPERPARAMETER OPTIMIZATION

Current federated learning methods struggle in cases with heterogeneous client-side data distributions which can quickly lead to divergent local models and a collapse in performance. Careful hyper-parameter tuning is particularly important in these cases. Hyper-parameters can be optimized using gradient descent to minimize the final validation loss (Maclaurin et al., 2015; Franceschi et al., 2017). Moreover, hyper-parameters can be optimized based on reinforcement learning methods (Guo et al., 2022; Jomaa et al., 2019; Mostafa, 2019). However, in this paper, optimization aggregation weights is not our main novelty. Instead, we focus on leveraging this toolbox on our well-designed but unexplored scenarios and examining the crucial training dynamics in FL in a principled way.

### B.4    MOST RELEVANT WORKS TO FEDAWO

We notice that two related works also optimize the aggregation weight (AW) by gradient descent. The first is AUTO-FEDAVG (Xia et al., 2021), which optimizes AW on different institutional medical data to realize personalized medicine. AUTO-FEDAVG adopts Softmax and Dirichlet functions as the base functions in optimizing AW. The second is a decentralized FL algorithm called L2C (Li et al., 2022a). It adopts a peer-to-peer (P2P) communication protocol and uses the local dataset of each client to optimize the collaborative weights with other clients. L2C assumes that different clients have various learning tasks, so it learns an adaptive weight for personalized collaborative learning. However, these two works all assume a normalized AW, whose $L_1$ norm equals 1, so they did not devise the global weight shrinking strategy for training a more generalized global model. Also, they are for specific application scenarios, like medical AI or P2P FL, and they did not introduce the aggregation weight optimization method to general FL and understand the training dynamics from

the learned AW. Additionally, these previous methods are all about personalization while we focus on generalization from the global perspective.

## C MORE RESULTS AND ANALYSES

### C.1 CLIENT COHERENCE

**The relationship with gradient diversity.** The conclusion of gradient diversity (Yin et al., 2018) is opposite to the one of gradient coherence. Gradient diversity argues that higher similarities between workers' gradients will degrade performance in distributed mini-batch SGD, while gradient coherence claims that higher similarities between the gradients of samples will boost generalization (Yin et al., 2018; Chatterjee, 2019). Moreover, gradient diversity is somewhat controversial. As argued in the line of works about gradient coherence (Chatterjee & Zielinski, 2020; Chatterjee, 2019), the manuscript of gradient diversity did not explicitly measure the gradient diversity in the experiments (or further study its properties): only experiments on CIFAR-10 can be found where they replicate $1/r$ of the dataset $r$ times and show that greater the value of r less the effectiveness of mini-batching to speed up. Apart from this controversy, the strongly-convex assumption in the theorem of gradient diversity (Yin et al., 2018) may make it weaker to generalize its conclusions in neural networks while we are studying the empirical properties in FL with neural networks. Taking the above statements into consideration, gradient diversity may be infeasible in our settings.

**The relationship with gradient diversity.** There are some works (Karimireddy et al., 2020; Li et al., 2020b) taking the bounded gradient dissimilarity assumption to deduce theorems. In their assumptions, they bound the gradient sum or gradient norm, but we use the cosine similarity to study how the clients interplay with each other and contribute to the global. So the perspectives are quite different. Additionally, there are previous works [] in FL that use cosine similarity of clients' gradients to improve personalization, however, we focus on the training dynamics in generalization, and one of our novel findings is we discover a critical point exists and the periods that before or after this point play different roles in the global generalization.

**Heterogeneity also affects the optimal aggregation weight.** We set up a three-node toy example on CIFAR-10 by hybrid Dirichlet sampling as shown in Figure 11. We first sample client 0's data distribution by Dirichlet sampling according to $\alpha_1$, then we sample data distributions for clients 1 and 2 on the remaining data with $\alpha_2$. We set up three settings with different $\alpha_1, \alpha_2$ and illustrate the data distributions on the **Left column** in Figure 11. In the example, the aggregation weights (AWs) are $[\lambda_0, \lambda_1, \lambda_2]$, we regularize the weights as $\lambda_0 + \lambda_0 + \lambda_0 = 1$ which is a plane that can be visualized in 2-D. We uniformly sample points on the plane to obtain global models with different AW and compute the test loss, and then the loss landscapes on the plane can be visualized. We implement FEDAVG for 100 rounds and record the loss landscape and the optimal weight on the loss landscape in each round, then we illustrate the loss landscape of round 10 on the **Middle column** and the optimal weight trajectory on the **Right column** of Figure 11.

In these settings, clients have different heterogeneity degrees: in the first setting, client 0 has a balanced dataset while the data of clients 1 and 2 are complementary; in the second and third settings, clients 1 and 2 have the same data distribution, which differs from the client 0's. From Figure 11, it is evident that the weight of FEDAVG is biased from optimal weights when heterogeneity degrees vary in clients, we can draw the following conclusions: (1) optimal weight can be viewed as a Gaussian distribution in the aggregation weight hyperplane; (2) the mean of the Gaussian will drift towards to the directions where data are more inter-heterogeneous (for instance, in the third setting, client 0's major classes are 2, 3 and 8 while client 1 and 2 have rare data on these classes, so client 0's contribution is more dominant); (3) the variance of the Gaussian is larger in inter-homogeneous direction and is smaller in inter-heterogeneous direction (the variance along the client 1-client 2 direction is large in the second and third settings, because the two clients have inter-homogeneous data; opposite phenomenon is shown in the first setting, where client 1 and 2 have inter-heterogeneous data); (4) the flatness of loss landscape on aggregation weight hyperplane is consistent with the variance of the Gaussian, which means the directions with more significant variance will have flatter curvature in the landscape. From our analysis, it is clear that clients' contributions to the global model should not be solely measured by dataset size, and the heterogeneity degree should also be taken into account. And we observe that in a more heterogeneous environment, the loss landscape is sharper, which means the bias from optimal weight will cause more generalization drop. In other words, in a heterogeneous environment, appropriate aggregation weight matters more.

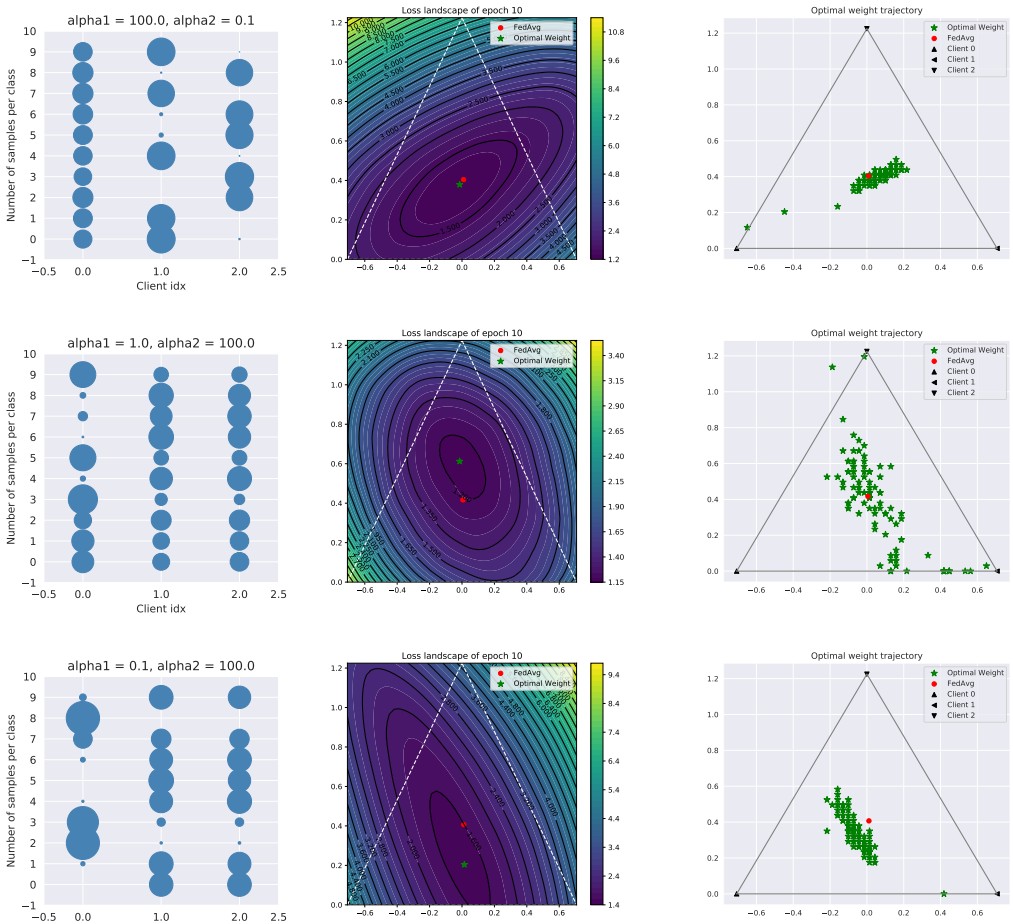

Figure 11: **Heterogeneity also affects the optimal aggregation weight.** A three-node toy example on CIFAR-10 is shown. **Left:** Data distribution of each client, note that each client has the same dataset size. **Middle:** Loss landscape on the plane of aggregation weight, it is noticed that FEDAVG is off the optimal and the landscape has various flatness in different directions. **Right:** optimal weight trajectory during training, the optimal weights are biased from FEDAVG.

**Visualization of the hybrid NonIID setting of Figure 1.** We visualize the hybrid NonIID setting of Figure 1 in Figure 12. We take $\alpha_1 = 10$ and $\alpha_2 = 0.1$, so the first 10 clients (indexed 0-9) have class-balanced data while the last 10 clients (indexed 10-19) have class-imbalanced data.

| Factors | $E = 1$ | $E = 5$ | $E = 1 \& E = 5$ |
|---|---|---|---|
| Dataset size (DS) | -0.098 | **0.21** | 0.035 |
| Heterogeneity degree (HD) | **0.41** | 0.024 | 0.35 |
| DS×HD | 0.26 | 0.17 | **0.31** |

Table 6: **Pearson correlation coefficient analysis of AW.** Heterogeneity degree is calculated as the reciprocal of the variance of class distribution for each client. We the accumulated weights during the training as clients' AW.

**Data size or heterogeneity? A correlation analysis.** Data size and heterogeneity all affects clients' contributions to the global model, but which affects it most? As in previous literature, the importance is depicted by the dataset size that clients with more data will be assigned larger weights. According to the analysis in Figure 11, the importance of weight may be associated with the heterogeneity degrees of clients. To explore which factor is more dominant in the AW optimized by attentive AWO,

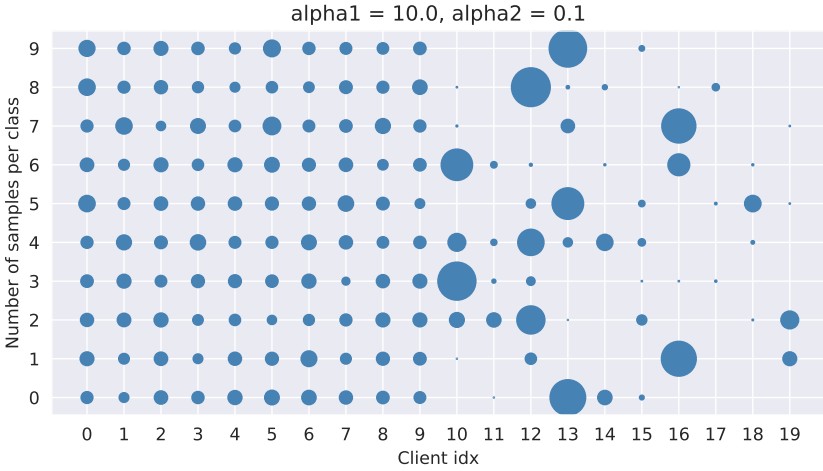

Figure 12: **Data distribution of Figure 1**

Table 7: **More results about fixed $\gamma$ across different architectures in various NonIID settings.**

| $\gamma$ | 1.0 | 0.99 | 0.97 | 0.95 | 0.93 | 0.9 |
|---|---|---|---|---|---|---|
| Model | | | $\alpha = 10$ | | | |
| SimpleCNN | 65.96 | 67.19 | 69.41 | **69.81** | 69.69 | 69.59 |
| AlexNet | 73.9 | 74.43 | 74.96 | 75.12 | **75.33** | 74.06 |
| ResNet8 | 76.26 | 75.63 | 76.92 | **77.23** | 76.9 | 76.61 |
| Model | | | $\alpha = 0.5$ | | | |
| SimpleCNN | 65.78 | 66.59 | 67.93 | **68.85** | 68.75 | 68.25 |
| AlexNet | 73.72 | 73.06 | 73.89 | **73.98** | 73.6 | 73.33 |
| ResNet8 | 73.4 | 73.93 | **75.39** | 74.12 | 73.66 | 73.46 |
| Model | | | $\alpha = 0.2$ | | | |
| SimpleCNN | 63.52 | 64.68 | 63.72 | **65.82** | 65.4 | 64.97 |
| AlexNet | 68.41 | 70.46 | **70.87** | 70.74 | 70.58 | 69.42 |
| ResNet8 | 71.85 | 70.96 | **72.76** | 72.04 | 71.25 | 62.32 |
| Model | | | $\alpha = 0.1$ | | | |
| SimpleCNN | 60.57 | 61.22 | 61.83 | **62.05** | 62.05 | 60.85 |
| AlexNet | **66.18** | 65.25 | 64.74 | 64.23 | 64.16 | 61.24 |
| ResNet8 | **63.89** | 60.55 | 61.38 | 59.23 | 58.76 | 39.85 |

we have made a Pearson correlation coefficient analysis in Table 6. Results show that dataset size is more dominant when the local epoch is large; otherwise, the heterogeneity degree. This phenomenon is intuitive: when the local epoch increases, clients with a larger dataset will have more local iterations than others (Wang et al., 2020b), so their updates are more dominant. In the cases where the local epoch is small, clients' updates are of similar volumes, here the updates' directions are much more important since balanced clients are prone to have stronger coherence, and their AWs are larger in model aggregation. We combine two factors by multiplication, and the result shows that the combined indicator is more dominant when the two cases are mixed.

## C.2 GLOBAL WEIGHT SHRINKING

**Fixed $\gamma$.** We add more results about global weight shrinking experiments with fixed $\gamma$ as in Table 7. It is found that when data are more NonIID, fixed $\gamma$ will cause negative effects; this is more dominant when $\alpha = 0.1$ and the models are AlexNet or ResNet8.
**Adaptive GWS with global learning rate.** We conduct experiments with the adaptive GWS under different global learning rates for both IID and NonIID settings. We train SimpleCNN on CIFAR10

Table 8: The performance of adaptive GWS under different global learning rates.

| Global learning rate | IID ($\alpha = 100$) | | | NonIID ($\alpha = 1$) | | |
| --- | --- | --- | --- | --- | --- | --- |
| | 0.5 | 1 | 1.5 | 0.5 | 1 | 1.5 |
| FedAvg | 69.15 | 68.18 | 64.35 | 68.71 | 67.09 | 64.00 |
| Adaptive GWS | 71.45 | 71.98 | 71.13 | 69.65 | 71.02 | 71.04 |
| $\gamma$ of Adaptive GWS | 0.986 | 0.974 | 0.963 | 0.991 | 0.979 | 0.967 |

with 1 local epoch, and the results are reported in Table 8. It can be observed that in both IID and NonIID settings, a small global server learning rate can improve FedAvg's performance. In contrast, the larger global learning rate, the smaller the learned $\gamma$ (stronger regularization). It is aligned with our insights in the main paper that larger pseudo gradients require stronger regularization. Moreover, Adaptive GWS is robust to the choice of the global server learning rate, especially in IID setting.

**Adaptive GWS under various heterogeneity.** We show adaptive GWS works under various heterogeneity and visualize $\gamma$ and the norm of the global gradient in each setting, as in Figure 13. It demonstrates that adaptive GWS can boost performance under different NonIID settings, but it has smaller benefit when the system is extremely NonIID (i.e., $\alpha = 0.1$). Additionally, according to the Right figure of Figure 13, except for the outlier $\gamma$ when $\alpha = 10$, the learned $\gamma$ decreases when data become more IID, causing stronger weight shrinking effect. We think this is a result of a balance between optimization and regularization. The volumes of global gradients change when the heterogeneity changes. The norm of global gradient increases when data become more IID, and it requires smaller $\gamma$ to cause stronger regularization.

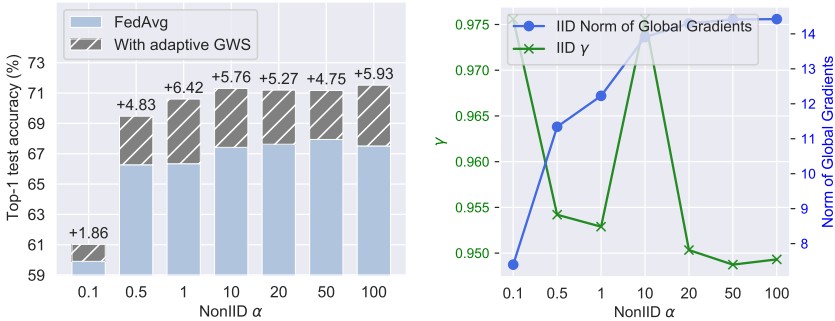

Figure 13: **Adaptive GWS under various heterogeneity. Left:** Test accuracy gains with adaptive GWS. In all settings, adaptive GWS can bring performance gains. **Right:** Learned $\gamma$ of adaptive GWS in different settings. $\gamma$ decreases when data become more IID, causing the stronger weight shrinking effect. This is due to the changes in the volumes of global gradients. The norm of global gradient increases when data become more IID, and it requires smaller $\gamma$ to cause stronger regularization.

**More results of general understanding of adaptive GWS.** First, we first visualize the norm of model parameter weight during training as in the Left figure of Figure 14. Adaptive GWS results in a smaller model parameter during training. Second, we use two common metrics to measure the flatness of loss landscape during training as in the Middle and Right figures of Figure 14, and they are the hessian eigenvalue based metrics. The dominant hessian eigenvalue evaluates the worst-case loss landscape, which means the larger top 1 eigenvalue indicates the greater change in the loss along this direction and the sharper the minima (Keskar et al., 2017). We adopt the top 1 hessian eigenvalue and the ratio of top 1 and top 5, which are commonly used as a proxy for flatness (Jastrzebski et al., 2020; Fort & Jastrzebski, 2019). Usually, a smaller top 1 hessian eigenvalue and a smaller ratio of top 1 hessian eigenvalue and top 5 indicates flatter curvature of NN. As in the figures, during the training, FEDAVG generates global models with sharp landscapes whereas adaptive GWS tends to generate more generalized models with flatter curvatures.

**The distribution of $r$.** We visualize $r$ values of all experiments in Figure 5 and Figure 13 as in Figure 15. It is found that the distribution of $r$ can be approximated into a Gaussian distribution with mean around 20.5.

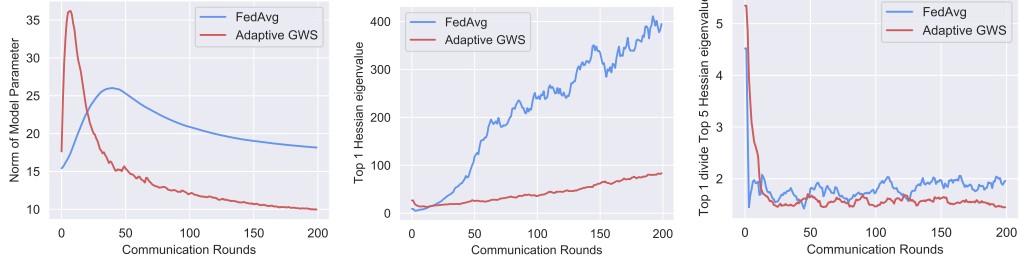

Figure 14: **More results of general understanding of adaptive GWS. Left:** Adaptive GWS results in a smaller model parameter during training. **Middle:** Smaller top 1 hessian eigenvalue indicates flatter curvature of NN. The result shows FEDAVG tends to generate sharper global models during training while adaptive GWS seeks flatter networks. **Right:** The ratio of top 1 hessian eigenvalue and top 5 is another indicator, a smaller value means flatter minima.

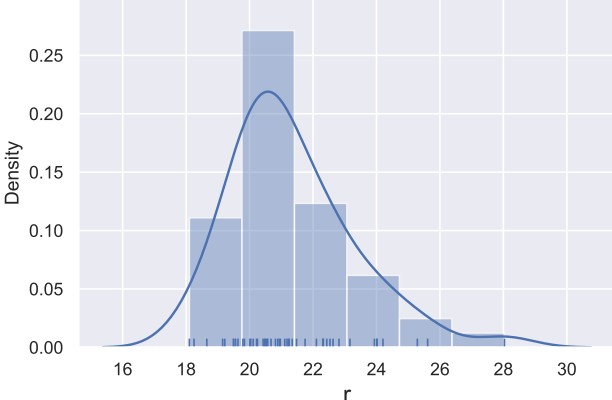

Figure 15: **Distribution of $r$.** We visualize $r$ values of all experiments in Figure 5 and Figure 13 and find that the distribution of $r$ can be approximated into a Gaussian distribution with mean around 20.5.

### C.3 COMAPRED EXPERIMENTS OF FEDAWO

We add the test accuracy curves to show the learning processes of the algorithms and visualize them in Figure 16 (FasionMNIST), Figure 17 (CIFAR-10), and Figure 18 (CIFAR-100). The curves are according to the results in Table 5. It shows that FEDAWO surpasses the baseline algorithms in most cases. Besides, FEDAWO is steady in the learning curves and it avoids over-fitting in the late training.

### C.4 DISTRIBUTION SHIFT OF PROXY DATASET

One may think it is a strong assumption that the distribution of the proxy dataset is the same as the global distribution. To reflect the superiority of FedAWO, we consider two challenging scenarios. We train ResNet20 on CIFAR10 and set the number of local epochs to 1 for both of them in IID and NonIID.

Table 9: The performance of the distribution shift scenario 1 where the clients' data are overall long-tailed ($\rho = 5$) and the proxy data are balanced.

| Method | IID ($\alpha = 100$) | NonIID ($\alpha = 1$) |
|---|---|---|
| FedAvg | 61.12 | 59.82 |
| FedDF | 39.03 | 40.68 |
| FedAWO | **67.61** | **66.48** |

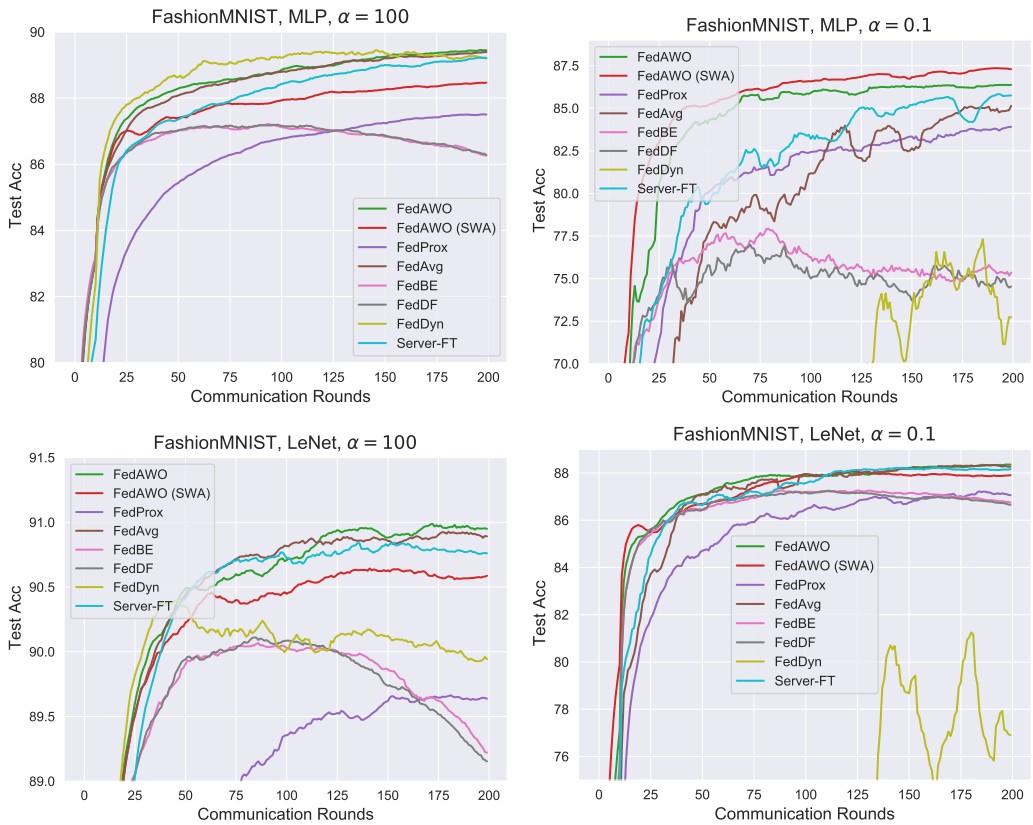

Figure 16: **Test accuracy curves of algorithms under FashionMNIST.** According to the results in Table 5.

Table 10: The performance of the distribution shift scenario 2 where the clients' data are overall balanced and the proxy data are long-tailed ($\rho = 10$).

| Setting | IID ($\alpha = 100$) | | | NonIID ($\alpha = 1$) | | |
|---|---|---|---|---|---|---|
| Type of proxy data | Balanced | Long-tailed | | Balanced | Long-tailed | |
| Balanced Sampling | - | w/o | w | - | w/o | w |
| FedAvg | 75.24 | 75.24 | 75.24 | 73.46 | 73.46 | **73.46** |
| FedDF | 76.20 | 74.04 | 73.31 | 74.39 | 73.99 | 73.14 |
| FedAWO | **79.40** | **77.14** | **78.56** | **76.70** | **76.78** | 70.42 |

- **Scenario 1: The clients' data are overall long-tailed while the proxy data are balanced.** The results are reported in Table 9, which illustrates that our method FedAWO also performs well. In comparison, the server-side ensemble distillation method FedDF has poor results in this setting, and it is worse than FedAvg.
- **Scenario 2: The clients' data are overall balanced while the proxy data are long-tailed.** The results are reported in Table 10, and we also find our method FedAWO can improve the performance. Due to the long-tailed distribution of the proxy dataset, we additionally design one extra strategy: balanced sampling, which means that we first sample the long-tailed proxy data into a smaller but balanced dataset and then use it. In the IID setting, the balanced sampling method can improve FedAWO's optimization. In contrast, in the NonIID setting, the original long-tailed data work well for FedAWO, and the balanced sampling even reduces the accuracy. Moreover, there is an interesting finding that the balance degree of the proxy dataset is more critical in IID settings. In comparison, FedDF has inferior results when the proxy set is long-tailed, and our FedAWO can improve generalization over FedAvg in both IID and NonIID scenarios.

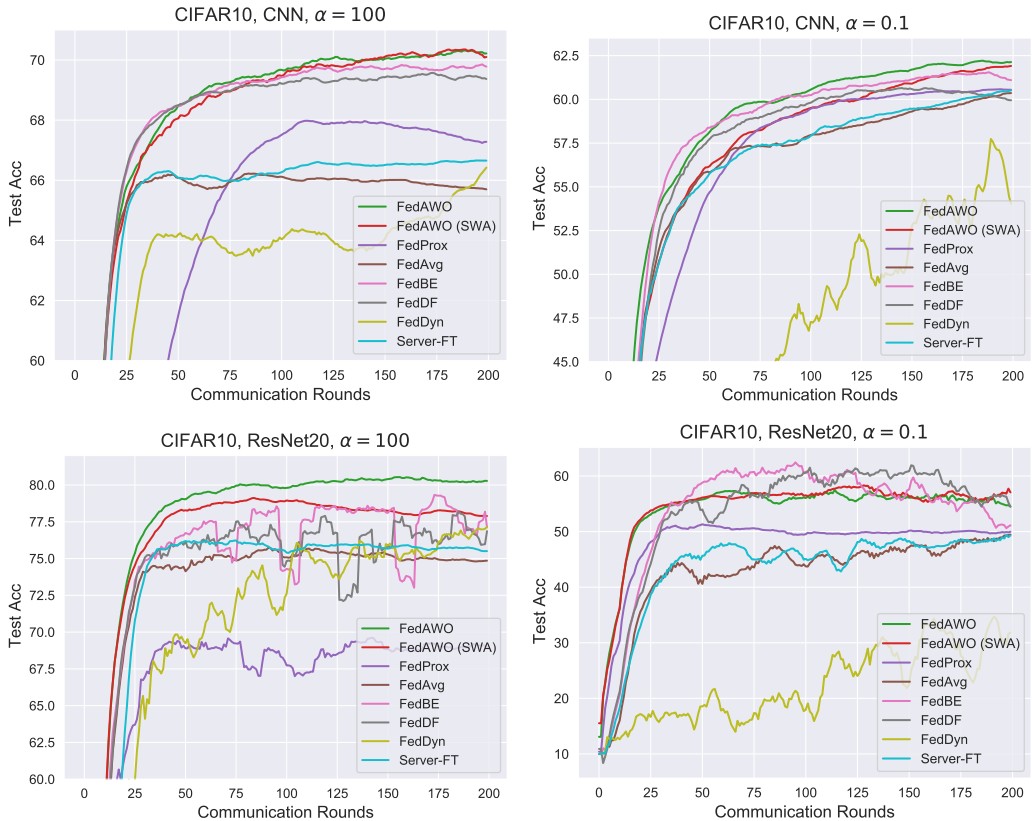

Figure 17: **Test accuracy curves of algorithms under CIFAR10.** According to the results in Table 5.

## D    ADDITIONAL DETAILS OF FEDAWO

In FEDAWO, we optimize AW on the server as Equation 6, and there are constraints that $\lambda_i \geq 0, \|\boldsymbol{\lambda}\|_1 = 1$. To realize these constraints, we adopt base functions in $\lambda$, and there are two alternatives, the quadratic function and the exponential function.

$$\text{Quadratic: } \lambda_i = \frac{x_i^2}{\sum_j^N x_j^2}; \text{Exponential: } \lambda_i = \frac{e^{x_i}}{\sum_j^N e^{x_j}}. \tag{9}$$

$\boldsymbol{x}$ is the variable that determines the value of $\boldsymbol{\lambda}$. We compute the gradients of $\boldsymbol{x}$ to updates $\boldsymbol{\lambda}$. By using the base functions, $\boldsymbol{\lambda}$ can meet the constraints of non-negativity and $L_1 = 1$. The exponential function is the same as the Softmax function and we find these two functions have similar performances overall, so we only adopt the exponential function in the experiments.

## E    IMPLEMENTATION DETAILS

### E.1    ENVIRONMENT.

We conduct experiments under Python 3.8.5 and Pytorch 1.12.0. We use 4 Quadro RTX 8000 GPUs for computation.

### E.2    DATA

**Data partition.** To generate NonIID data partition amongst clients, we use Dirichlet distribution sampling in the trainset of each dataset. In our implementation, apart from clients having different class distributions, clients also have different dataset sizes; we think this partition is more realistic in practical scenarios. For the data partition in Figure 1 and Figure 11, we use a hybrid Dirichlet sampling to generate a FL system with both class-balanced clients and class-imbalanced clients. Specifically, we first generate all-client distribution with $\alpha_1$, and we only keep half of these clients.

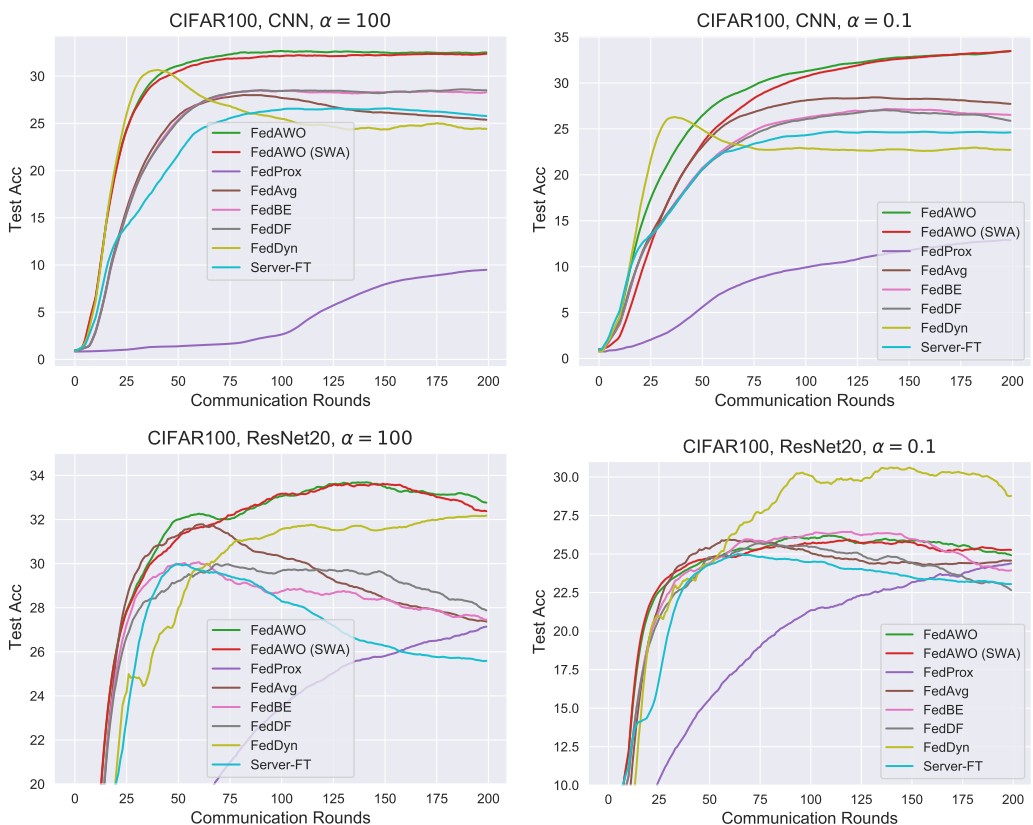

Figure 18: **Test accuracy curves of algorithms under CIFAR100.** According to the results in Table 5.

Then we use the remaining data to generate the distribution of remaining clients with $\alpha_2$. For the data in Figure 1, we first generate a 20-client distribution with $\alpha_1 = 10$ and keep the first 10 clients as the balanced clients, then we use the remaining data to generate distribution of the last 10 imbalanced clients with $\alpha_2 = 0.1$. The distribution is shown in Figure 12.

**Data augmentation.** We adopt no data augmentation in the experiments.

**Proxy dataset.** We use a small and class-balanced proxy dataset on the server. In Table 5, we use proxy datasets with 10 samples per class, which means, for FashionMNIST and CIFAR-10, there are 100 samples in the proxy datasets, and for CIFAR-100, there are 1000 samples in the proxy datasets. The proxy datasets are randomly selected from the testset of each dataset. Then we use the remaining data in the testset to test the global models' performance for all compared methods. For Table 3 and Figure 10, we use CIFAR-10 and a 100-sample proxy dataset, while in Table 4, we use CIFAR-10 and a 1000-sample proxy dataset.

### E.3 MODEL

**SimpleCNN and MLP.** The SimpleCNN for CIFAR-10 and CIFAR-100 is a convolution neural network model with ReLU activations which consists of 3 convolutional layers followed by 2 fully connected layers. The first convolutional layer is of size (3, 32, 3) followed by a max pooling layer of size (2, 2). The second and third convolutional layers are of sizes (32, 64, 3) and (64, 64, 3), respectively. The last two connected layers are of sizes (64*4*4, 64) and (64, num_classes, respectively. The MLP model for FasionMNIST is a three-layer MLP model with ReLU activations. The first layer is of size (28*28, 200), the second is of size (200, 200), and the last is (200, 10).

**ResNet and DenseNet.** We followed the model architectures used in (Li et al., 2018). The numbers of the model names mean the number of layers of the models. Naturally, the larger number indicates a deeper network. For WRN56_4 in Table 4, it is an abbreviation of Wide-ResNet56-4, where "4" refers to four times as many filters per layer.

### E.4 RANDOMNESS

Randomness is very important in a fair comparison. In all experiments throughout the paper, we all implement the experiments three times with different random seeds and report the averaged results. We use random seeds 8, 9 and 10 in all experiments. Given a random seed, we set torch, numpy, and random functions as the same random seed to make the data partitions and other settings identical. To make sure all algorithms have the same initial model, we save an initial model for each architecture and load the saved initial model at the beginning of one experiment. Also, for the experiments with partial participation, the participating clients in each round are vital in determining the model performance, and to guarantee fairness, we save the sequences of participating clients in each round and load the sequences in all experiments. This will make sure that, given a random seed and participation ratio, every algorithm will have the same sampled clients in each round.

### E.5 EVALUATION

We evaluate the global model performance on the testset of each dataset. The testset is mostly class-balanced and can reflect the global learning objective of an FL system. Therefore, we reckon the performance of the model on the testset can indicate the generalization performance of global models. In all experiments, we run 200 rounds and take the average test accuracy of the last 10 rounds as the final test accuracy for each experiment. For the indicators during training in section 4, like $\gamma$, $r$, the norm of global gradient, and the norm of GWS pseudo-gradient, we take the averaged values in the middle stage of training, that is the average of 90-110 rounds.

### E.6 HYPERPARAMETER

**Learning rate and the scheduler.** We set the initial learning rates (LR) as 0.08 in CIFAR-10 and FashionMNIST and set LR as 0.01 in CIFAR-100. We set a decaying LR scheduler in all experiments; that is, in each round, the local LR is 0.99*(LR of the last round).

**Local weight decay.** We adopt local weight decay in all experiments. For CIFAR-10 and FashionMNIST, we set the weight decay factor as 5e-4, and for CIFAR-100, we set it as 5e-5.

**Optimizer.** We set SGD optimizer as the clients' local solver and set momentum as 0.9. For the server-side optimizer (FEDDF, FEDBE, SERVER-FT, and FEDAWO), we use Adam optimizer and betas=(0.5, 0.999).

**Hyperparameter for FL algorithms.** For FEDDF, FEDBE and FEDAWO, we set the server epoch as 100. We observe for SERVER-FT, this epoch is too large that it will cause negative effects, so we set the epoch as 2 for SERVER-FT. We set $\mu = 0.001$ in FEDPROX and $\alpha = 0.01$ in FEDDYN as suggested in their official implementations or papers. For FEDBE, we use the Gaussian mode in SWAG server. We did not use temperature smoothing in the ensemble distillation methods FEDDF and FEDBE.

