# OpenReview forum: "Understanding the Training Dynamics in Federated Deep Learning via Aggregation Weight Optimization"
_ICLR.cc/2023/Conference — Submitted to ICLR 2023_

### Official Review · Reviewer_eJUT · 2022-10-24

**Confidence:** 4
**Correctness:** 3
**Technical Novelty And Significance:** 2
**Empirical Novelty And Significance:** 2
**Recommendation:** 5

**Clarity, Quality, Novelty And Reproducibility:**

The paper is well-written and easy to follow. However, I think most of the findings are not new or surprising for a reader with experience in FL.

**Strength And Weaknesses:**

Strength:
1. The paper is well-organized and easy to follow.
2. The empirical improvement is significant.

Weaknesses/questiones:
1. How is the proxy dataset selected? Proxy dataset does not seem to be a common practice in most FL settings, and it is not fair to compare it with other methods not using proxy dataset.
2. How is Eq(3) and Eq(4) optimized? The default aggregation weight in FL is used in Eq(4), instead of the proposed $\lambda$. Why not jointly optimize $\gamma$ and $\lambda$?
3. In Eq(2), why is the first order Tyler expansion $-\eta\langle g^t,g^t\rangle$? Shouldn't it be $-\eta\langle g^t, \nabla \mathcal{L}(w^t)\rangle$?
4. After Eq(2), why do you assume all gradients have the same norm? This does not seem to be realistic.
5. Regarding cosine similarity/gradient coherence, it is not a new concept. For example, check [1] (there are also many other works). Moreover, it is actually the core problem in FL studied by many prior works as it contributes to the client drift issue.
6. The global weight shrinking does not seem novel to me. It is applying weight decay to the pseudo-gradient (local update of a client) and I do not think it is new.
7. How do you compare GWS with server learning rate both theoretically and empirically?
8. How do you compare methods employing server optimizers, which apply SGD/Adam variants to pseudo-gradient/local update from clients, both theoretically and empirically?
9. Optimizing aggregation weights is not new. There has been more advanced methods to deal with it, e.g. reinforcement learning [2,3]. While the method proposed is very simple by using a proxy dataset.

References:
[1] Yin, Dong, et al. "Gradient diversity: a key ingredient for scalable distributed learning." International Conference on Artificial Intelligence and Statistics. PMLR, 2018.
[2] Auto-FedRL: Federated Hyperparameter Optimization for Multi-institutional Medical Image Segmentation. ECCV 2022.
[3] Robust federated learning through representation matching and adaptive hyper-parameters. arXiv preprint arXiv:1912.13075 (2019)

**Summary Of The Paper:**

The author proposes to optimize the aggregation weight of each client, and find that the aggregation weight is closely related to gradient coherence. The proposed method improves the heterogeneity coherence. The author also proposes global weight shrinking to improve the training performance. Moreover, the author makes many definitions of new concepts and try to explain how the training dynamic are affected in FL. Empirical experiments show improvement over existing methods.

**Summary Of The Review:**

The author made a lot of new definitions and propose to optimize the aggregation weight and gradient weight shrinking. However, the use of proxy dataset make it less practically appealing, and the proposed techniques do not seem novel. Most of the findings are not new to me, too.

---

> ### Author Response · Authors · 2022-11-13
> **Response to Reviewer eJUT (4/4)**
>
>
> > 8. Response to "How do you compare methods employing server optimizers, which apply SGD/Adam variants to pseudo-gradient/local update from clients, both theoretically and empirically?"
>
> Thanks for the comment. To address the concern, we included some comparisons with FedAdam [4] in the revised manuscript in both IID and NonIID settings (training SimpleCNN on CIFAR10 with 1 local epoch).
>
> Generally, we find that FedAdam cannot outperform FedAvg in our implementation and its performance largely relies on the hyperparameters. On the contrary, our method is more "adaptive" than the Adam variant, as we are hyper-parameter tuning free and can learn the weights automatically.
>
>
> Table X6: The performance of compared methods in both IID and NonIID.
> |Method|IID ($\alpha=100$）|NonIID ($\alpha=1$）|
> | ------------------------ | -------- | -------- |
> |FedAvg|68.02 |67.25 |
> |FedAdam ($\eta_g$=0.001)|67.20 |66.20 |
> |FedAdam ($\eta_g$=0.01)|63.34 |62.69 |
> |FedAWO|72.00 |71.20 |
>
>
>
>
> > 9. Response to "Optimizing aggregation weights is not new. There has been more advanced methods to deal with it, e.g. reinforcement learning [2,3]. While the method proposed is very simple by using a proxy dataset."
>
> Thanks for the recommendation about the related works, and we have included them in the related works part of Appendix B.3 on page 15 in the new version.
>
> We would like to clarify that
> * Optimization aggregation weights is not our main novelty. Instead, we focus on leveraging this toolbox on our well-designed but unexplored scenarios and explaining the crucial training dynamics in FL in a principled way.
> * FedAWO is a side-product and only serves to justify the importance and effectiveness of our empirical insights and understandings.
> * We believe such understanding is very crucial to the whole community, similar to the seminal understanding work [5, 6, 7, 8]. To avoid potential misunderstanding, we have rewritten the sentences in the revised draft (blued text on page 2).
> * RL-based papers pointed out by the reviewer recommended are interesting, and they developed complicated methods for server-side optimization. However, they are less relevant to the main claims/contributions of our paper.
>
> ### Reference
> [1] Chatterjee S. Coherent Gradients: An Approach to Understanding Generalization in Gradient Descent-based Optimization[C]//International Conference on Learning Representations. 2019.
>
> [2] Yin D, Pananjady A, Lam M, et al. Gradient diversity: a key ingredient for scalable distributed learning[C]//International Conference on Artificial Intelligence and Statistics. PMLR, 2018: 1998-2007.
>
> [3] Chatterjee S, Zielinski P. Making Coherence Out of Nothing At All: Measuring Evolution of Gradient Alignment[J]. 2020.
>
> [4] Reddi S J, Charles Z, Zaheer M, et al. Adaptive Federated Optimization[C]//International Conference on Learning Representations. 2020.
>
> [5] Zhang C, Bengio S, Hardt M, et al. Understanding deep learning (still) requires rethinking generalization[J]. Communications of the ACM, 2021, 64(3): 107-115.
>
> [6] Arpit D, Jastrzębski S, Ballas N, et al. A closer look at memorization in deep networks[C]//International conference on machine learning. PMLR, 2017: 233-242.
>
> [7] Lewkowycz A, Gur-Ari G. On the training dynamics of deep networks with $ L_2 $ regularization[J]. Advances in Neural Information Processing Systems, 2020, 33: 4790-4799.
>
> [8] Zhang G, Wang C, Xu B, et al. Three Mechanisms of Weight Decay Regularization[C]//International Conference on Learning Representations. 2018.

---

> ### Author Response · Authors · 2022-11-13
> **Response to Reviewer eJUT (3/4)**
>
>
> > 6. Response to "The global weight shrinking does not seem novel to me. It is applying weight decay to the pseudo-gradient (local update of a client) and I do not think it is new."
>
> We agree with the reviewer that the concept of weight decay is not new to mini-batch SGD. However, rather than introducing this concept as our novelty, our main contributions lie in (1) understanding how much will this global weight shrinking affect the training dynamics in FL, and (2) its distinct insights compared to that in mini-batch SGD. In details,
> 1. Weight decay in mini-batch SGD is favorable to small factors (e.g. $10^{-5}$), while global weight shrinking in FL prefers to use a value at a relatively large magnitude of $10^{-1}$. Such difference may dramatically change the training dynamics, attributing to the key difference between mini-batch SGD and FL.
> 2. In contrast to our global weight shrinkage, which is an adaptive approach that learns an optimal regularization from the data, the value of weight decay in mini-batch SGD is often determined prior to the training. We believe it may inspire some interesting directions on adaptive regularization, even for mini-batch SGD.
>
> > 7. Response to "How do you compare GWS with server learning rate both theoretically and empirically?"
>
> Thanks for the comment. To address the concern, we have included some additional numerical results in the revised manuscript in Appendix C.2 on page 18, in terms of the adaptive GWS under different global learning rates, for both IID and NonIID settings (training SimpleCNN on CIFAR10 with 1 local epoch).
>
> As a first step to understanding the training dynamics in FL, our empirical work is a crucial yet solid step in the field. Some insights:
> 1. In both IID and NonIID settings, a small global server learning rate can improve FedAvg's performance.
> 2. Adaptive GWS is robust to the choice of the global server learning rate, especially in IID setting.
> 3. The larger global learning rate, the smaller the learned $\gamma$ (stronger regularization). It is aligned with our insights in the main paper that larger pseudo gradients require stronger regularization.
>
>
> Table X4: The performance of adaptive GWS under different global learning rates in IID ($\alpha=100$)
> |Global learning rate|0.5|1|1.5|
> | ------------------------ | -------- | -------- |-------- |
> |FedAvg|69.15 |68.18 |64.35 |
> |Adaptive GWS|71.45 |71.98 |71.13 |
> |$\gamma$ of Adaptive GWS|0.986|0.974|0.963|
>
> Table X5: The performance of adaptive GWS under different global learning rates in NonIID ($\alpha=1$)
> |Global learning rate|0.5|1|1.5|
> | ------------------------ | -------- | -------- |-------- |
> |FedAvg|68.71 |67.09 |64.00 |
> |Adaptive GWS|69.65 |71.02 |71.04 |
> |$\gamma$ of Adaptive GWS|0.991|0.979|0.967|

---

> ### Author Response · Authors · 2022-11-13
> **Response to Reviewer eJUT (2/4)**
>
>
> > 4. Response to "After Eq(2), why do you assume all gradients have the same norm? This does not seem to be realistic."
>
> Sorry for the confusion. In fact, it is a simplified statement where we use a bounded gradient norm assumption. We have clarified this point in the revised manuscript (blued text on page 4).
>
>
> > 5. Response to "Regarding cosine similarity/gradient coherence, it is not a new concept. For example, check [1] (there are also many other works). Moreover, it is actually the core problem in FL studied by many prior works as it contributes to the client drift issue."
>
> Thank you for pointing that out. We also noticed these concepts, e.g., cosine similarity/gradient diversity, at the initial stage of research. However, it is orthogonal to our insights and contributions (as acknowledged by reviewer q3AP). We would like to clarify as following.
> * Gradient diversity
>     1. The conclusion of gradient diversity is opposite to the one of gradient coherence. Gradient diversity argues that higher similarities between workers' gradients will degrade performance in distributed mini-batch SGD, while gradient coherence claims that higher similarities between the gradients of samples will boost generalization [1, 2].
>     2. Gradient diversity is somewhat controversial. As argued in the line of works about gradient coherence [1, 3], the manuscript on gradient diversity did not explicitly measure the gradient diversity in the experiments (or further study its properties): only experiments on CIFAR-10 can be found where they replicate $1/r$ of the dataset $r$ times and show that greater the value of r less the effectiveness of mini-batching to speed up.
>     3. Apart from this controversy, the strongly-convex assumption in the theorem of gradient diversity [2] may make it weaker to generalize its conclusions in neural networks while we are studying the empirical properties in FL with neural networks.
>     4. Taking the above statements into consideration, we reckon gradient diversity may not be a feasible choice in our settings.
> * Gradient similarity
>     1. Despite that many prior works, e.g., SCAFFOLD and FedProx, take the bounded gradient dissimilarity assumption to deduce the convergence theorems, they rely on the bounded gradient sum or gradient norm per update step. We instead leverage the cosine similarity to understand the training dynamics in FL and study how the clients interplay with each other. The key insights derived from us cannot be captured by these existing convergence analyses (our contribution therein).
>     2. Though there may exist many prior works in FL to use cosine similarity of clients' gradients to improve personalization, we focus on understanding the training dynamics and their impact on the generalization performance. As one of our novel findings, we identify the existence of a critical point, where the periods before or after this point play different roles in global generalization.
>
> Given that these discussions could further aid the reader in identifying the contribution of our paper, we have added it in Appendix C.1 on page 15 of the revised paper.

---

> ### Author Response · Authors · 2022-11-13
> **Response to Reviewer eJUT (1/4)**
>
> We thank the reviewer for the acknowledgements that our paper is well-organized and the empirical improvement is significant. We kindly address your questions as follows.
>
> > 1. Response to "How is the proxy dataset selected? Proxy dataset does not seem to be a common practice in most FL settings, and it is not fair to compare it with other methods not using proxy dataset."
>
> Thanks for the comment. We would like to clarify that the proxy dataset is just a tool: *by leveraging a proxy dataset through the lens of aggregation weight optimization*, we are able to deliver our main contributions on *the understanding and insights in the training dynamics of federated learning*.
>
> For a fair evaluation, we have already compared our FedAWO---as a side product of our understandings---with other SOTA proxy-data-based FL methods, e.g., FedBE, FedDF, and Server-FT. Some related statements and empirical results can be found in the second paragraph on page 9 of the initial submission.
>
> We also elaborate on our implementation details below regarding the selection of the proxy dataset in FedAWO:
> * In our implementation, we just randomly sampled a small and balanced dataset from the testset, and we excluded the proxy set when validation.
> * The proxy dataset is so small that it generally contains 10 samples per class in CIFAR10. We reckon the small proxy dataset is realistic and attainable in practice.
> * The distribution of proxy data can be robust to that of the training data. For more details, please see [Reply to Reviewer q3AP Q1](https://openreview.net/forum?id=VJjtgKzrmj&noteId=xlYfz8WgokN), Section 5 on page 9 and Appendix C.4 on page 20 in the revised manuscript.
>
> > 2. Response to "How is Eq(3) and Eq(4) optimized? The default aggregation weight in FL is used in Eq(4), instead of the proposed λ. Why not jointly optimize $\gamma$ and $\lambda$?"
>
> Thanks for the comment. We would like to clarify the intuition behind our choice regarding fixing $\gamma$ to optimize $\lambda$ in Section 3 while fixing $\lambda$ to optimize $\gamma$ in Section 4:
> 1. The main contribution lies in understanding the training dynamics in FL, rather than focusing on proposing a new method. Due to the different roles of $\gamma$ and $\lambda$ in the training dynamics, we need to control variables so as to identify the impact of one variable. More precisely,
>     * optimizing $\lambda$ alone helps us to understand client coherence that how the clients interplay with each other and contribute to the global objective;
>     * optimizing $\gamma$ alone helps us to understand the optimal global weight shrinking about the balance between regularization and optimization and how heterogeneity and local epochs affect that balance.
> 2. It also explains the question raised by the reviewer, i.e., "The default aggregation weight in FL is used in Eq(4) instead of the proposed $\lambda$". Note that in Eq(4), we fix the default client aggregation weight $\lambda$ as FedAvg and optimize $\gamma$. In Eq(3), we set the default data-size-based $\lambda$ as the optimization starting point and then optimize $\lambda$.
> 3. In our implementation, we optimize $\gamma$ or $\lambda$ by gradient descent on the proxy dataset, while fixing the previous model parameters, the global learning rate, and the clients' pseudo-gradient. In the rebuttal, we have included our code as a supplementary file to ease the understanding.
>
> > 3. Response to "In Eq(2), why is the first order Taylor expansion $−\eta⟨g_t,g_t⟩$? Shouldn't it be $−\eta⟨g_t, \nabla L(w_t)⟩$?"
>
> Sorry for the missing details. We agree with the reviewer that the first order Taylor expansion is $−\eta\langle g_t, \nabla L(w_t) \rangle$. Following the treatment in Eq(1) of the original paper of gradient coherence [1], we also made an approximation---by using $\approx$ instead of $=$ in our original submission---between the derivative of the global loss function $\nabla L(w_t)$ and the computed global pseudo-gradient $g_t$.

---

> ### Author Response · Authors · 2022-11-17
> **Hoping that our response could address your concerns**
>
> Dear Reviewer eJUT,
>
> Thank you again for reading our rebuttal. We have faithfully given detailed responses to your concerns. We would appreciate it if you could let us know if our response has sufficiently addressed your questions and thus kindly reconsider your score. Thank you.
>
> Best wishes,
>
> Authors

---

> ### Author Response · Authors · 2022-11-27
> **To Reviewer eJUT**
>
> Dear Reviewer eJUT,
>
> Thank you again for your detailed comments. In the rebuttal, we have clarified some misunderstandings and added additional experiments.
>
> * _**Clarify misunderstandings.**_
>     * Importantly, we have emphasized that **our main contributions are the understandings of FL's training dynamics** by taking aggregation weight optimization as a tool, and we are not claiming aggregation weight optimization is our initiative.
>     * To obtain the understandings, for $\gamma$ and $\lambda$, **we control variables by fixing one while optimizing another in Eq(3) and (4)**.
>     * We have clarified that **it is totally fair** that most compared baselines are using a proxy dataset.
>     * We have made a clarification on the significance and novelty of adaptive global weight shrinking.
>     * We have discussed the differences between our paper and the existing works in gradient diversity/similarity.
> * _**Include additional experiments.**_
>     * We have added experiments about our method compared/combined with **server learning rate** and **server optimizer methods**.
>
> After the rebuttal, we hope all the concerns are relieved, and some misunderstandings are peacefully erased.
>
> We sincerely invite the reviewer to take the precious time to check our response and give the kind feedback. We will also be much delighted if the reviewer could reconsider the evaluation.
>
> Thanks again for your attention and patience.
>
> Best wishes,
>
> Authors

---

### Official Review · Reviewer_q3AP · 2022-10-25

**Confidence:** 4
**Correctness:** 3
**Technical Novelty And Significance:** 3
**Empirical Novelty And Significance:** 3
**Recommendation:** 6

**Clarity, Quality, Novelty And Reproducibility:**

The paper is very well written and clear to understand. The paper brings forth a lot of new insights in terms of the training dynamics of FL. However, a lot of benefits of the proposed algorithm seems to come from the usage of proxy data at the server which has been used before in other papers.

**Strength And Weaknesses:**

Strengths:
+ The paper is well written and the problem under consideration is well motivated. The proposed algorithm is light weight, i.e., doesn't add any computational overhead for the devices but instead adds it to the server which makes the algorithm attractive to use. The algorithm is easy to use and doesn't add any additional hyperparameters.
+ The authors provide various new insights in terms of the effects of gradient coherence and heterogeneous coherence and its interplay with global weight shrinking and critical points to better explain the training dynamics in FL.
+ Experimental results are comprehensive in terms of different datasets, models, different levels of heterogeneity both in terms of non-iidness and the number of samples on clients.

Weaknesses:
- It seems success of the approach is tied to the proxy dataset being a representative sample of the entire data distribution. No matter how small or how large the proxy dataset is, the server gets access to the true data distribution of all the clients. This is a major limitation of the approach. Having a proxy dataset at the server is something entirely possible, but the distribution of it being represenative of the true data distribution is a strong assumption. It is not clear how the performance of the algorithm is affected, when the proxy dataset is not distributionally similar to the data distribution of all the clients as a whole. It would especially strengthen the paper, as to how much of the benefit is attributable to the distribution of the proxy dataset and how much of it is from adaptively optimizing for $\gamma$ and $\lambda_{i}$'s.
- The proposed approach is devoid of any theoretical guarantees and is based on observations based on SGD being the local solver. In particular, the observations concerning gradient coherence wouldn't carry over if each client undergoes multiple local steps.
- The experiments are missing some details regarding the number of local epochs considered for different baselines and the performance of FedAWO in settings involving multiple local steps.

**Summary Of The Paper:**

This paper studies the problem of optimizing for aggregation weights assigned to client model updates along with weight decay so as to aid generalization. In particular, the authors take the viewpoint of seeing the aggregation and server side update analogous to that of mini-batch SGD in centralized training and propose an algorithm named FedAWO. In particular, the proposed algorithm optimizes for weight decay and aggregation weights by making use of a proxy dataset at the server. Experiments on different datasets and models demonstrate the efficacy of the approach.

**Summary Of The Review:**

The paper solves a well motivated problem with a simple, practical and well examined solution. The proposed algorithms introduces no additional hyperparameters which makes the solution particularly effective. However, the benefits claimed seems to be tightly tied to the similarity of the proxy dataset with that of the sum total data distribution of the clients. The paper could be strengthened by studying the robustness of the approach in terms of benefits still intact even when the data at the server is distributionally different from the data distribution of the sum total of all clients.

---

> ### Author Response · Authors · 2022-11-13
> **Response to Reviewer q3AP (2/2)**
>
> > 2. Response to "The proposed approach is devoid of any theoretical guarantees and is based on observations based on SGD being the local solver. In particular, the observations concerning gradient coherence wouldn't carry over if each client undergoes multiple local steps."
>
> Starting from a theoretical analytic framework, our paper focuses on providing insights from extensive empirical aspects to better interpret the training dynamics in FL. We believe it is crucial to the FL community and could be similar to the seminal understanding works in other fields of deep learning [1, 2, 3, 4].
>
> We leverage the analogy between mini-batch SGD and FL (or equivalently local SGD) and develop our principled aspects and understandings.
> In details:
> * The observations concerning gradient coherence of mini-batch SGD could be carried over to local SGD by extending the coherence between samples to the coherence between clients, in which the stochastic gradient or the sum of stochastic gradients are evaluated.
> * The analogy from an FL perspective allows the identification of some specific insights of FL, including the critical point in the client coherence and the optimal global weight shrinking factor exploration. These insights can inspire future theoretical investigations, but theories are beyond the scope of this paper.
>
> > 3. Response to "The experiments are missing some details regarding the number of local epochs considered for different baselines and the performance of FedAWO in settings involving multiple local steps."
>
> During the revision, we have added the missing descriptions on the number of local epochs (E) or the heterogeneity degree ($\alpha$) in each table. The readers can check the added details in the table's caption.
>
> ### Reference
> [1] Zhang C, Bengio S, Hardt M, et al. Understanding deep learning (still) requires rethinking generalization[J]. Communications of the ACM, 2021, 64(3): 107-115.
>
> [2] Arpit D, Jastrzębski S, Ballas N, et al. A closer look at memorization in deep networks[C]//International conference on machine learning. PMLR, 2017: 233-242.
>
> [3] Lewkowycz A, Gur-Ari G. On the training dynamics of deep networks with $ L_2 $ regularization[J]. Advances in Neural Information Processing Systems, 2020, 33: 4790-4799.
>
> [4] Zhang G, Wang C, Xu B, et al. Three Mechanisms of Weight Decay Regularization[C]//International Conference on Learning Representations. 2018.

---

> > ### Comment · Reviewer_q3AP · 2022-12-06
> > **Thanks for your rebuttal**
> >
> > Thanks a lot for a thorough rebuttal and clarifying that FedAWO is not an overarching contribution of this paper. While I am convinced that server side data helps, in practice, however it is not possible to know whether server's data is balance and whether the distribution of data over the clients is IID or non-IID. Because of this, it is not possible to preemptively decide what's the best way to retain data on the server. It needs more exploration and should be quantified in terms of a suitably designed metric to quantify the difference in distributions of the data at the server and the sum total of the clients.
> >
> > Having said that, I have decided to raise my score.

---

> ### Author Response · Authors · 2022-11-13
> **Response to Reviewer q3AP (1/2)**
>
> We thank the reviewer for recognizing our paper is well-written, well-motivated, and provides various new insights.
>
> Following the reviewer's suggestions, we have added additional experiments in the revision to make FedAWO more solid, though we would like to kindly note that our main contribution is the crucial understanding of FL training dynamics rather than FedAWO itself. The detailed responses are listed as follows.
>
> > 1. Response to "It seems success of the approach is tied to the proxy dataset being a representative sample of the entire data distribution. No matter how small or how large the proxy dataset is, the server gets access to the true data distribution of all the clients. This is a major limitation of the approach. Having a proxy dataset at the server is something entirely possible, but the distribution of it being represenative of the true data distribution is a strong assumption. It is not clear how the performance of the algorithm is affected, when the proxy dataset is not distributionally similar to the data distribution of all the clients as a whole. It would especially strengthen the paper, as to how much of the benefit is attributable to the distribution of the proxy dataset and how much of it is from adaptively optimizing for $\gamma$ and $\ lambda$'s."
>
> Though our main contribution lies in providing some crucial insights into the training dynamics of FL, we added some complementary experiments in the revised manuscript (Section 5 on page 9 and Appendix C.4 on page 20) and below to justify that FedAWO still works when there exists a distribution shift between the proxy dataset and the global data distribution of clients.
>
> **To answer the key concern regarding the effects caused by the proxy dataset distributions**, two challenging scenarios are considered: (1) the clients' data are overall long-tailed while the proxy data are balanced; (2) the proxy data are long-tailed while the clients' data are overall balanced. We train ResNet20 on CIFAR10 and set the number of local epochs to 1. The results are shown in the tables below.
>
> We can find that:
> 1. For the global long-tailed data and the balanced proxy data setting, our method also performs well. In comparison, the server-side ensemble distillation method FedDF has poor results in this setting, and it is worse than FedAvg.
> 2. In the setting where the proxy dataset is long-tailed and the whole training data are balanced, we also find FedAWO can improve the performance. Note that
>     * Two extra strategies were taken to deal with the long-tailed proxy set, namely (1) we directly use it, and (2) we can sample the long-tailed proxy data into a smaller but balanced dataset (balanced sampling).
>     * *These two strategies will have different effects in IID or NonIID settings*. In the IID setting, the balanced sampling method can improve FedAWO's optimization. In contrast, in the NonIID setting, the original long-tailed data work well for FedAWO, and the balanced sampling even reduces the accuracy.
>     * This is an interesting finding that the balance degree of the proxy dataset is more critical in IID settings. In comparison, FedDF has inferior results when the proxy set is long-tailed, and our FedAWO can improve generalization over FedAvg in both IID and NonIID scenarios.
>
>
> Table X1: The performance of the distribution shift scenario where the clients' data are overall long-tailed ($\rho=5$) and the proxy data are balanced.
> | Method  | IID($\alpha=100$)  | NonIID($\alpha=1$)    |
> | ------------------------ | -------- | -------- |
> |FedAvg|61.12 |59.82 |
> |FedDF|39.03 |40.68 |
> |FedAWO|67.61 |66.48 |
>
> Table X2: The performance of the distribution shift scenario where the clients' data are overall balanced and IID ($\alpha=100$) and the proxy data are long-tailed ($\rho=10$).
>
> |Type of proxy data|Balanced|Long-tailed|Long-tailed|
> | ------------------------ | -------- | -------- |--------|
> |Balanced Sampling|-|w/o|w|
> |FedAvg|75.24 |75.24 |75.24 |
> |FedDF|76.20 |74.04 |73.31 |
> |FedAWO|79.40 |77.14 |78.56 |
>
> Table X3: The performance of the distribution shift scenario where the clients' data are overall balanced and NonIID ($\alpha=1$) and the proxy data are long-tailed ($\rho=10$).
> |Type of proxy data|Balanced|Long-tailed|Long-tailed|
> | ------------------------ | -------- | -------- |--------|
> |Balanced Sampling|-|w/o|w|
> |FedAvg|73.46 |73.46 |73.46 |
> |FedDF|74.39 |73.99 |73.14 |
> |FedAWO|76.70 |76.78 |70.42 |

---

> ### Author Response · Authors · 2022-11-17
> **Hoping that our response could address your concerns**
>
> Dear Reviewer q3AP,
>
> Thank you again for reading our rebuttal. We have faithfully given detailed responses to your concerns. We would appreciate it if you could let us know if our response has sufficiently addressed your questions and thus kindly reconsider your score. Thank you.
>
> Best wishes,
>
> Authors

---

> ### Author Response · Authors · 2022-11-27
> **To Reviewer q3AP**
>
> Dear Reviewer q3AP,
>
> Thank you again for your constructive comments. Since your concerns are mainly about the proposed FedAWO, in the rebuttal, we have clarified **our main contributions are the understandings of FL's training dynamics**, and FedAWO is a simple side product of these understandings.
>
> * However, we have also added detailed experiments to relieve your concerns about FedAWO. Essentially, we have shown in the experiments that **FedAWO is robust and effective when the distribution shift between the proxy dataset and the whole training data exists**.
>
> * In addition, your questions about local gradient coherence and setting details are also thoroughly answered.
>
> We believe our paper is more solid after the rebuttal. We will appreciate that if you would consider raising the score according to our revision.
>
> Thank you for your precious time.
>
> Best wishes,
>
> Authors

---

### Official Review · Reviewer_dT3M · 2022-10-31

**Confidence:** 2
**Correctness:** 3
**Technical Novelty And Significance:** 2
**Empirical Novelty And Significance:** 3
**Recommendation:** 6

**Clarity, Quality, Novelty And Reproducibility:**

Overall, this paper is well written and clearly presented.

More materials should be provided to help evaluate novelty (see Strength And Weaknesses).

The code should be made public to validate reproducibility owing to the fact that implementation details are missing in the text.

**Strength And Weaknesses:**

Overall, this paper is well written and clearly presented.

The core components (client coherence and global weight shrinking regularization) of the proposed framework are carefully analyzed independently to each other before presenting the resulting FedAWO.

Related work on federated learning is partially missing; it is necessary to move key related work to the main text. As a result, the paper is not appropriately positioned in literature in a certain sense: Is it possible to introduce more aggregation parameters on the server side in a similar adaptive manner, except for the proposed $\gamma$ and $\lambda$? Are there any previous work adopting a similar adaptive aggregation approach?

The concept of “number of local epochs” is confusing. Intuitively, it is about how $g^t_i$ is derived locally for each client. However, since the preliminary about FL is not introduced in detail, it is not sufficiently readable.

**Summary Of The Paper:**

This paper investigates the training dynamics in federated deep learning from the perspective of mini-batch SGD. Client coherence and global weight shrinking regularization are introduced in the proposed FedAWO framework. The corresponding parameters are learned in the server side with a proxy dataset. FedAWO’s effectiveness is demonstrated empirically across several datasets and model architectures. Interestingly, FedAWO shows the ability to filter out corrupted clients.

**Summary Of The Review:**

See “Strength And Weaknesses” and “Clarity, Quality, Novelty And Reproducibility”.

---

> ### Author Response · Authors · 2022-11-13
> **Response to Reviewer dT3M**
>
> We thank the reviewer for the valuable comments, and we are delighted to hear from the reviewer that our paper is well-written and clearly presented. We give a detailed response as follows.
>
> > 1. Response to "Related work on federated learning is partially missing; it is necessary to move key related work to the main text."
>
> Thank you for the suggestion. Due to the space limits, we only briefly discussed the related work on our key concepts in Section 2 on page 3 using remarks (weight decay and gradient coherence). We have moved the most relevant related works from Appendix A.3 of the initial submission to the main paper (the first blued paragraph in section 1 on page 2) regarding the aggregation weight optimization in FL.
>
> Note that the previous works have several limitations.
> * They all assume normalized aggregation weights of clients' models (i.e. $\gamma=1$ in Equation 1), failing to identify the crucial aspect of the adaptive global weight shrinking.
> * They do not dive deeper to understand the FL's dynamics from the learned weights (our contributions therein).
>
>
> > 2. Response to "Is it possible to introduce more aggregation parameters on the server side in a similar adaptive manner, except for the proposed $\gamma$ and $\lambda$? Are there any previous work adopting a similar adaptive aggregation approach?"
>
> We note $\gamma$ and $\lambda$ are the most important aggregation weights to understand the training dynamics in terms of client coherence and global weight shrinking. Other aggregation weights, like the global learning rate $\eta_g$, are also of interest, but due to the focus and page length of the paper, we would rather study it in future work. About the previous works, we have answered in Response 1.
>
> > 3. Response to "The concept of "number of local epochs" is confusing. Intuitively, it is about how $g_i^t$ is derived locally for each client. However, since the preliminary about FL is not introduced in detail, it is not sufficiently readable."
>
> Many thanks for the suggestions for improving our readability to a broader audience. In the revision, we have added the preliminary about FL in Appendix A on page 14; such background knowledge is also referred in the main text as footnote 1 on page 2.
>
> Regarding the explanation of "the number of local epochs", we further added a sentence in the paragraph under Equation 1, justifying that it is about how $g_i^t$ is derived locally for each client.
>
>
> > 4. Response to "The code should be made public to validate reproducibility owing to the fact that implementation details are missing in the text."
>
> We have included the draft code in the newly updated supplementary materials, and we will release the official clean code upon acceptance.
>
> Note that due to page limits, we included the implementation details in Appendix D in our initial submission (current Appendix E). To improve the reproducibility, in the revision, we have added the key experimental settings about the number of local epochs or NonIID degrees in the captions of the main-text Tables.

---

> ### Author Response · Authors · 2022-11-17
> **Hoping that our response could address your concerns**
>
> Dear Reviewer dT3M,
>
> Thank you again for reading our rebuttal. We have faithfully given detailed responses to your concerns. We would appreciate it if you could let us know if our response has sufficiently addressed your questions and thus kindly reconsider your score. Thank you.
>
> Best wishes,
>
> Authors

---

> ### Author Response · Authors · 2022-11-27
> **To Reviewer dT3M**
>
> Dear Reviewer dT3M,
>
> We have fully addressed your concerns in the rebuttal and revision.
>
> Specifically, **we have added the preliminary of FL and provided our code** to improve readability and reproducibility.
>
> Thank you again for your valuable comments. It would be very kind of you to check our response and reconsider the evaluation.
>
> We will be very glad to hear your feedback.
>
> Best wishes,
>
> Authors

---

### Author Response · Authors · 2022-11-13
**General Response**

We thank the reviewers for their valuable comments. We are glad that the reviewers found our paper is well-written and easy to follow (Reviewers dT3M, q3AP, eJUT), our insights are new (Reviewer q3AP), the proposed FedAWO is attractive and easy to use (Reviewer q3AP), and the empirical improvement of FedAWO is significant (Reviewers dT3M,  eJUT).

### Clarifying the contributions
Most of the comments from the reviewers are on our finally proposed method FedAWO.
We would like to clarify that *our main contributions are the understanding and insights into the training dynamics of federated learning through the lens of aggregation weight optimization*. Our proposed FedAWO---though being simple yet efficient---is a side product to showcase the significance of these understandings as well as its potential practical impact; we believe these insights are non-trivial and would definitely inspire other crucial inventions in the field.

We provided responses to each reviewer separately below, and we will incorporate all the feedback in the final version.

---

### Decision · Program_Chairs · 2023-01-20

**Decision:**

Reject

**Justification For Why Not Higher Score:**

I believe the work does not raise to the bar required of an ICLR paper. Explanation included in my metareview.

**Justification For Why Not Lower Score:**

N/A

**Metareview: Summary, Strengths And Weaknesses:**

This paper investigates the training dynamics in federated deep learning from the perspective of mini-batch SGD. Client coherence and global weight shrinking regularization are introduced in the proposed FedAWO framework. The corresponding parameters are learned in the server side with a proxy dataset. FedAWO’s effectiveness is demonstrated empirically across several datasets and model architectures. Interestingly, FedAWO shows the ability to filter out corrupted clients.

Issues include:

1. As an expert in FL, I find it difficult to appreciate the research question posed by the authors. I believe it is fundamentally misguided - the ideas look ad hoc, and are a combination of ideas that already appeared in the literature (often without citing the original or prior sources). In summary, I do not see much novelty here.

2. Further, I do not believe that local SGD should be studied using the lens of minibatch SGD; the methods are fundementaly different. Recent results studying the nature of local SGD theoretically have shown that local steps can be seen as a communication acceleration mechanism (without any homogeneity assumptions). This is something that minibatch SGD cannot do. Similarly, recent results from a different line of works have shown that local steps in SGD can be provably linked to generalization (with some structured homogeneity/similarity assumptions). This is unrelated to the phenomena the authors study here. In short, I do not believe the questions the authors are asking are worth asking. Moreover, they are not sufficiently well justified. Relevant recent literature which gives deep insights into what local SGD does is largely ignored by the authors. The mixture of FL related phenomena the authors study seems very ad-hoc from the point of view of were the literature on local SGD stands now. Large swaths of very relevant literature are omitted.

3. The paper contains no theoretical results. This is not required, of course, but empirical works should come with a particularly strong empirical component, which I do not see here.

4. It seems success of the approach is tied to the proxy dataset being a representative sample of the entire data distribution. This was criticized by some reviewers, and I concur.

5. I have read the positive reviews, and I am unconvinced by the strength of the arguments used to praise the paper. That is, I see a disconnect between the scores and the wording used to justify it.

In summary, I believe the work does not raise to the bar required of an ICLR paper.

AC


**Summary Of Ac-Reviewer Meeting:**

No meeting was held

---

> ### Author Response · Authors · 2023-02-08
> **Response to the Meta-review**
>
> Dear program committee,
>
> We thank the meta-reviewer for his/her time and valuable feedback. We would like to clarify the misunderstanding the meta-review might have below.
>
> Our main contributions—different from the extensive theoretical study focusing on analyzing federated learning through convergence theory—lie in understanding practical federated deep learning where many phenomena therein cannot be explained by theory:
>
> - **Weight decay and global weight shrinkage.** As an understanding paper for federated deep learning, we study the effect of global weight shrinkage in FL, an analogy effect of weight decay in mini-batch SGD. To the best of our knowledge, the training dynamics of mini-batch SGD induced with weight decay currently cannot be explained by the SGD convergence theory.
> - **Gradient coherence and client coherence.** As we detailed in the [rebuttal](https://openreview.net/forum?id=VJjtgKzrmj&noteId=4qvTlTN_3Hv), the existing notions like gradient diversity and gradient similarity are insufficient to precisely interpret the training dynamics in federated deep learning. Instead, similar to the empirical federated deep learning work published in NeurIPS 2021, titled On large-cohort training for federated learning, we use the analogy between mini-batch SGD and FL. Note that our contributions are orthogonal: they study whether the conclusions in batch size can be generalized to cohort size, while we instead study the weight decay and gradient coherence (they can both be revealed through aggregation weights in FL).
>
> Following this line, we would like to further claim some potentially misunderstood points:
>
> - **Novelty and Literature.**
>     - We would like to argue that [Novelty does not necessarily mean that the method has to be something completely new. Using existing method in a different way from existing results can also be novel, as long as the proposed method or theory makes sense, such as leading to significant performance improvement, new insights, or new understandings.](https://openreview.net/forum?id=3eQEil044E&noteId=3KNGuiNrHpm) Our novelty is also recognized by Reviewers dT3M and q3AP.
>     - We agree with the meta-reviewer that mini-batch SGD is not theoretically equivalent to local SGD from the perspective of convergence theory: neither our claim nor contribution. As an empirical understanding work for federated deep learning, we only leverage them—similar to the [NeurIPS 2021](https://arxiv.org/abs/2106.07820)—to motivate our understanding. We believe it may not be our responsibility to include the extensive literature on improving the convergence theory of local SGD, and we believe we have included all the most relevant works (whether orthogonal or not) in the main paper.
> - **Soundness.** Due to the space limitation, we cannot include our extensive results in each section of the main paper, but we have all referred to these results in the appendix. We believe our empirical results are solid enough to convey our empirical understanding.
> - **Concern** **on the proxy dataset** (indeed has already been solved in the rebuttal).
>     - As we claimed in our manuscript and our rebuttal (see [link #1](https://openreview.net/forum?id=VJjtgKzrmj&noteId=0rI6NeE0fT), [link #2](https://openreview.net/forum?id=VJjtgKzrmj&noteId=Mqk1F8X5MFw), and [link #3](https://openreview.net/forum?id=VJjtgKzrmj&noteId=LAdL-_ioG7w)), the proposed FedAWO is a side-product to showcase the significance of our empirical insights.
>     - We have conducted extensive [experiments](https://openreview.net/forum?id=VJjtgKzrmj&noteId=xlYfz8WgokN) in the rebuttal to address the concern; Reviewer q3AP has acknowledged this point and raised the score accordingly. More precisely,  FedAWO is still superior to other methods, when considering the distribution of the proxy dataset to be different from the entire data distribution in two scenarios, i.e., (1) server long-tailed, clients balanced; (2) server balanced, clients long-tailed.
>
> We thank all reviewers again for the valuable feedback. We have incorporated all of them to improve the clarity and soundness of our manuscript.